# Single atom engineering for radiotherapy-activated immune agonist prodrugs

Zexuan Ding[1,5], Xiaozhe Yin[2,5], Yuedan Zheng[3,5], Yiyan Li[4,5], Huanhuan Ge[1], Jianshu Feng[1], Ziyang Wang[3], Simiao Qiao[1], Qi Sun[4], Fashuo Yu[1], Zhanshan Hou[1], Yang-Xin Fu [1,2] ✉ & Zhibo Liu [1,3,4] ✉

Immunotherapy has revolutionized cancer treatment by leveraging the body's immune system to combat malignancies. However, on-target, off-tumour (OTOT) toxicity poses significant challenges, often leading to the failure of clinical trials for the development of immunotherapeutic drugs. The molecular engineering of clinically relevant, tumour-selective prodrugs, activated in a targeted way, could help minimize systemic toxicity while maximizing anti-tumour efficacy. Here, we propose a Single Atom Engineering for Radiotherapy-Activated Prodrug (SAE-RAP) technique for the development of radiotherapy-activatable small-molecule immune agonist prodrugs. We show that introducing a single oxygen atom into the TLR7/8 agonist R848 significantly reduces the $EC_{50}$ value by over 4000-fold, hence mitigating severe side effects following systemic administration. In preclinical tumour mouse models, exposure to radiotherapy removes the protective mask provided by the oxygen atom and locally rescues the activity of the prodrugs, triggering anti-tumour immunity and limiting the growth of primary and distal tumours. The SAE-RAP technique may be further utilized for developing radiotherapy-activated prodrugs for next-generation combination therapies that transcend traditional limitations.

Immunotherapy has revolutionized oncology by leveraging the innate power of the body's immune system to prevent, control, and even eliminate cancers[1]. However, a significant challenge in the development of immunotherapeutic drugs is the occurrence of on-target, off-tumour (OTOT) toxicity, including the potentially fatal "cytokine storm" triggered by potent drugs[2–4]. This issue has led to the failure of many clinical trials. Ideally, if these drugs could be molecularly engineered to be activity-blocked prodrugs, it might help reduce the uncontrollable OTOT toxicity during blood circulation. Then the prodrugs should be selectively activated in tumour sites to induce anti-tumour immunity. Such a prodrug engineering technique that addresses the above challenge has not yet been established.

Modern radiotherapy techniques, utilized by over 50% of cancer patients[5], can precisely target and eliminate tumour cells while triggering the production of tumour-associated antigens (TAA). Thus, radiotherapy has attracted considerable attention as a prospective approach for anti-tumour immunotherapy[6–10]. Unfortunately, the immunostimulatory effects of local radiotherapy in clinical treatments are limited, especially with the rare "abscopal effect"[11]. Considering the aforementioned situation, we wonder whether radiotherapy (ionizing radiations such as X-ray, γ-ray, etc.) can

[1]Changping Laboratory, Beijing, China. [2]Department of Basic Medical Sciences, School of Medicine, Tsinghua University, Beijing, China. [3]Beijing National Laboratory for Molecular Sciences, Radiochemistry and Radiation Chemistry Key Laboratory of Fundamental Science, Key Laboratory of Bioorganic Chemistry and Molecular Engineering of Ministry of Education, College of Chemistry and Molecular Engineering, Peking University, Beijing, China. [4]Peking University-Tsinghua University Centre for Life Sciences, Peking University, Beijing, China. [5]These authors contributed equally: Zexuan Ding, Xiaozhe Yin, Yuedan Zheng, Yiyan Li. ✉e-mail: yangxinfu@tsinghua.edu.cn; zbliu@pku.edu.cn

concurrently activate immunotherapeutic prodrugs in tumour tissues during treatments. In this scenario, radiotherapy has the potential to not only decrease tumour burden but also induce significant anti-tumour immunity capable of triggering potent "abscopal effects". Especially, radiotherapy offers the advantage of deep tissue penetration (up to 15 cm) for activating prodrugs in living bodies[12]. Hence, the radiotherapy-activated prodrug (RAP) strategy for developing immunotherapeutic prodrugs holds promise for enhancing the treatment efficacy and addressing systemic toxicity.

It is worth noting that small-molecule immunotherapeutic drugs offer multiple advantages, such as bioavailability, fast clearance, diverse target selection, and low-cost to potentially benefit a larger number of patients[13,14]. Among them, small-molecule immune agonists can effectively activate myeloid cells to present antigens and enhance T cell-mediated anti-tumour immunity[15–18]. Previously reported prodrug-based nanomedicines have contributed to the development of radiotherapy-activated immune prodrugs[19,20]. However, we aim to investigate a molecular engineering approach specifically for small-molecule prodrugs. In this study, toll-like receptor 7/8 (TLR7/8) agonists, predominantly in the clinical research phase, with only one approved for the treatment of skin diseases[21–23], were selected as model drugs. With the assistance of computer docking analysis, we discovered that active agonist molecules can be effectively blocked by the installation of a single oxygen atom. Furthermore, X-ray can strip the oxygen atom's protective mask, leading to prodrug activation, which is based on our previously reported radio-cleavage chemistry[24]. After systemic administration of prodrugs while mitigating severe toxicity, tumour-targeted local radiotherapy selectively unleashes the activity of immune agonists, thus triggering anti-tumour immunity for tumour treatment with the "abscopal effect" (Fig. 1a). We named this prodrug development technique Single Atom Engineering for Radiotherapy-Activated Prodrugs (SAE-RAP). Furthermore, we propose that the SAE-RAP technique could be used to develop prodrugs targeting alternative receptors. In sum, the SAE-RAP technique may provide a blueprint for safer and more effective prodrug design in next-generation combination therapy applications in clinics.

## Results

### Computer-aided single atom engineering for immune agonist R848

We chose a potent TLR7/8 agonist R848 as a model drug due to its considerable potential for cancer treatment[25]. The workflow for the computer-aided molecular engineering of R848 was designed as shown in Fig. 1b. The structure-activity relationship studies by docking analysis showed that the quinoline nitrogen atom and the C4 amine group synergistically form stable hydrogen bonds with Asp543 upon the binding of R848 to the interface of the TLR8 dimer. This interaction plays an essential role in receptor activation, which is consistent with a previous report[26]. Motivated by the "magic methyl effect"[27,28], we wonder whether a single oxygen atom engineered on the quinoline nitrogen atom (through a simple N-oxidation reaction) can block the immunostimulatory activity. To investigate the impact of the oxygen atom engineering, we synthesised the prodrug named O-R848 via one-step oxidation to install an oxygen atom at the quinoline nitrogen atom of the R848 (Supplementary Fig. 1 and Supplementary Figs. 28–30). Then, the docking analysis and kinetic simulations for the TLR8 dimer with R848 and O-R848 were conducted. The N-heterocyclic structure of R848 can form long-lasting and intricate interactions with the amino acid residues within the TLR8 pocket, particularly stabilizing TLR8 in the activated state through D543 and T574 interactions (Fig. 1b, Supplementary Fig. 2a and Supplementary Fig. 3a). Nevertheless, the installation of an oxygen atom in O-R848 disrupts the interaction between quinoline nitrogen and D543, thereby disrupting the intricate interaction network, leading to a shift in the positions of the two TLR8 monomers and preventing them from being locked in an activated state (Fig. 1b, Supplementary Fig. 2b and Supplementary Fig. 3b).

To evaluate whether a single oxygen atom can reduce the systemic OTOT toxicity caused by R848. Healthy C57BL/6 J mice were intravenously administered with O-R848 and R848 at a dose of 30 μmol/kg, respectively. After 4 h, it was observed that the production of proinflammatory cytokines in the serum induced by R848 was significantly elevated. Specifically, the levels of interferon-γ (IFN-γ) were 6.8 times higher, and those of tumour necrosis factor-α (TNF) were 37.9 times higher, in comparison to O-R848 (Fig. 1c). Subsequently, we compared the impact of systemic administration for 3 total times on the body weight. The data indicated that mice treated with R848 experienced significant side effects, resulting in a survival rate of less than 80%. In contrast, mice given O-R848 showed reduced weight loss (even less than 10% weight loss at a dosage of 90 μmol/kg), and no fatalities occurred (Fig. 1d). The results indicated that the oxygen atom engineering reduces the systemic toxicity and "cytokine storm" caused by R848.

Then, we conducted a RAW-Blue assay to assess the activity of O-R848 and R848. The results indicated that the median effect concentration (EC$_{50}$) of O-R848 was at least 4000-fold higher than that of R848 (Fig. 1e), suggesting that a single oxygen atom can block the immunostimulatory activity of R848. Through parallel artificial membrane permeability assay (PAMPA), the effective permeability coefficients (Pe) of R848 and O-R848 are $14.62 \pm 1.31$ and $0.43 \pm 0.09$ (Supplementary Table 1). This result means the cell permeability of R848 was ~34-fold reduced via single oxygen atom engineering, which may also contribute to the low toxicity observed for O-R848. It is noteworthy that the cytotoxicity experiments indicate that O-R848 incubated with MC38 and RAW264.7 cells induces lower cytotoxicity than R848 (Supplementary Fig. 4). This result also demonstrated that the effective concentrations for immune stimulation by R848 are markedly lower than their cytotoxic thresholds, suggesting that immune agonists mainly induce immune stimulation instead of directly causing cell death. Interestingly, X-ray-treated O-R848 exhibited restored immunostimulatory activity as that of R848 (Fig. 1f), indicating that ionizing radiation can rescue the blocked activity of oxygen atom-engineered R848.

### X-ray rescues the activity of oxygen atom-engineered immune agonist prodrugs

Subsequently, we investigated how X-ray rescues the activity of oxygen atom-engineered immune agonist prodrugs. Earlier studies have demonstrated that the active species produced by water radiolysis, including hydroxyl radicals and hydrated electrons (e$^-_{aq}$)[29,30]. In particular, the e$^-_{aq}$ was reported as a "super-reductant" for challenging reactions (Supplementary Fig. 5a, b)[31]. Consequently, we used O-R848 to verify whether X-ray-generated e$^-_{aq}$ can efficiently activate oxygen atom-engineered immune agonist prodrugs via reduction in aqueous solution (Fig. 2a and Supplementary Fig. 5c). The released R848 from O-R848 in PBS solutions were found in a radiation dose-dependent manner as detected by UPLC-MS (Fig. 2b and Supplementary Fig. 5d–f). The activation yield of R848 was approximately 120 nM/Gy, nearing the theoretical maximum. It is worth noting that the radiation reaction between e$^-_{aq}$ and O-R848 is nearly instant. Upon irradiation, the maximum release of R848 was detected immediately, without the need for additional waiting time (Fig. 2c and Supplementary Fig. 6). Importantly, this instant activation matches the short metabolism time of small-molecule drugs in living bodies. Then, in order to mechanistically verify that the reaction is indeed mediated by e$^-_{aq}$, O-R848 was irradiated either in acetonitrile solution or in phosphate buffered saline (PBS) solution with the e$^-_{aq}$ quencher KNO$_3$. The results revealed a significant decrease in the release yield of R848 compared to that in pure water and PBS solution (Supplementary Fig. 5g), indicating the dominance of e$^-_{aq}$ in the reaction. We subsequently investigated the

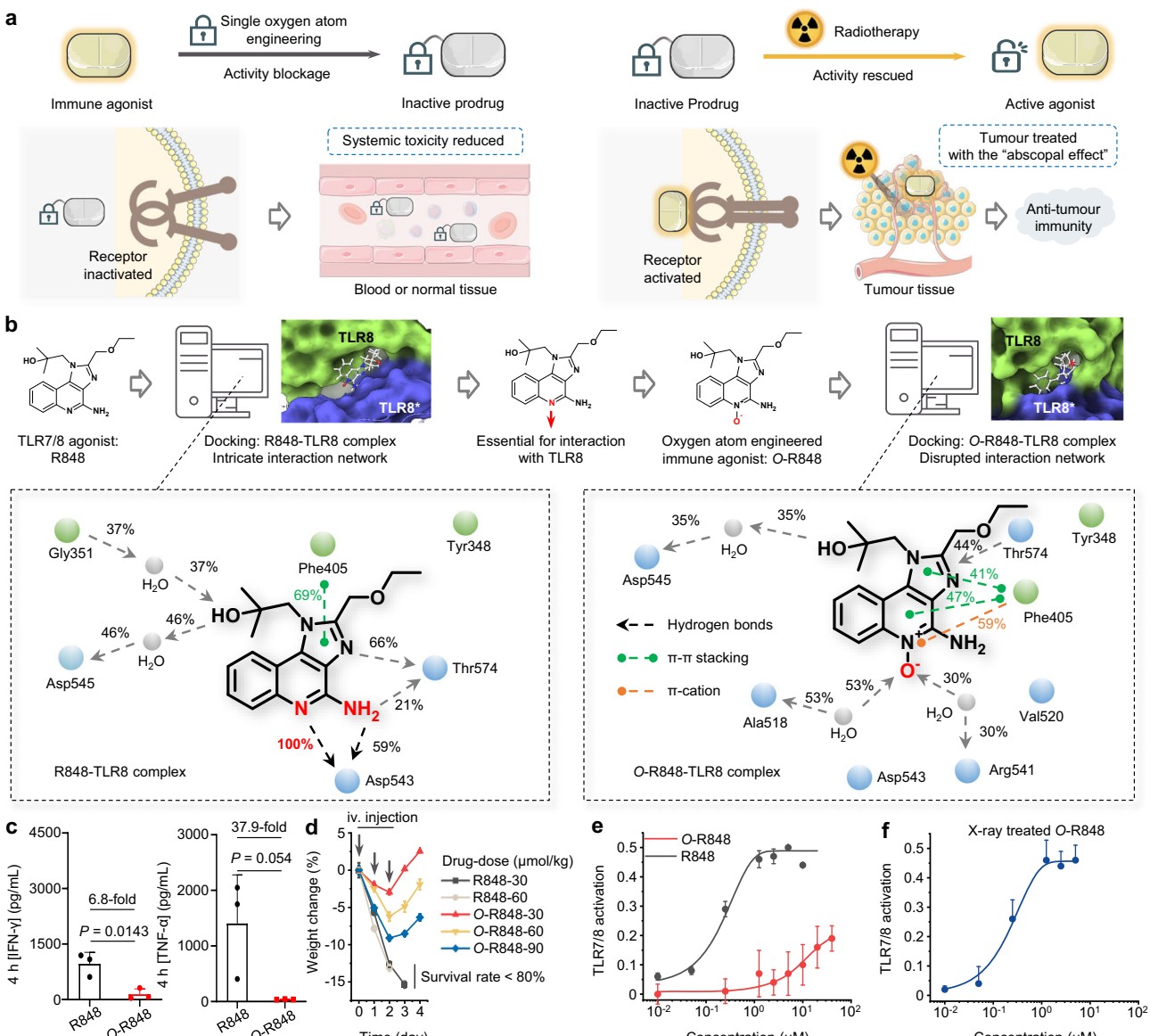

**Fig. 1 | Computer-aided single-atom engineering for small-molecule radio-therapy-activated immune agonist prodrugs. a** In this study, we propose a single oxygen atom engineering blockage approach to develop small-molecule immune agonist prodrugs for mitigating systemic on-target, off-tumour (OTOT) toxicity. The blockage of active drug molecules can be rescued in vivo by radiotherapy to treat tumours with the "abscopal effect". **b** Workflow of molecular engineering for immune agonist R848: Docking analysis (PDB: 3W3L) reveals that the quinoline nitrogen atom of R848 plays a key role in R848-TLR8 complex; Molecular engineering of R848 by a single oxygen atom (*O*-R848); Docking analysis reveals this oxygen atom can disrupt the intricate interaction network. **c** Serum IFN-γ and TNF

measurements taken from C57BL/6J mice (6-8-week-old, female) at 4 h post-intravenous injection of R848 and *O*-R848 at a dose of 30 μmol/kg, respectively. *n* = 3 mice for each group tested. **d** Body weight change measurements of C57BL/6J mice (6-8-week-old, female, *n* = 6 mice for each group) following 3 times intrave-nous administration of R848 and *O*-R848, respectively. Concentration-dependent TLR7/8 activation curves of R848, *O*-R848 (**e**), and *O*-R848 treated with X-ray (**f**) in RAW-Blue report cells after 24 h incubation (in **e**, **f**, *n* = 6 independently tested cell samples for each group). Data are presented as mean values ± s.d., two-tailed unpaired Student's *t*-test. Source data are provided as a Source Data file.

stability and reactivity of *O*-R848 within living cells. Following a 24-hour co-incubation of *O*-R848 with MC38 cells, there was no dis-cernible release of R848. However, when 10 μM *O*-R848 was exposed to 60 Gy X-ray irradiation, approximately 4.7 μM of R848 was promptly detected (Supplementary Fig. 5h). This data demonstrates that the *O*-R848 can remain stable in physiological environments and that radiotherapy activation reaction is biocompatible.

After demonstrating the efficient and instant activation of *O*-R848 by X-ray, we aim to validate the suitability of various commercial TLR7/8 agonists for this technique. Firstly, we synthesized 11 types of oxygen atom-engineered TLR7/8 agonist prodrugs, all of which contain imi-dazoquinoline (IMDQ) groups (Fig. 2d, Supplementary Fig. 1 and

Supplementary Figs. 31–60). The density functional theory (DFT) cal-culation employed the B3LYP method (detailed in the Supplementary Information) and the lowest unoccupied molecular orbital (LUMO) distribution of the representative molecules *O*-R848 (Fig. 2e) sug-gested that the π-system of the imidazoquinoline groups accom-modates the $e^-_{aq}$. Our previous work has illustrated that the reaction occurs when the absolute value of Gibbs free energy change ($|\Delta G|$) between oxygen atom-engineered prodrugs and its anion radical intermediate (after accommodating the $e^-_{aq}$) in the aqueous phase is greater than that of $e^-_{aq}$[30]. The $|\Delta G|$ values of different prodrugs com-pared with that of $e^-_{aq}$ suggested good reactivity (Fig. 2f and Supple-mentary Table 2). The experimental activation yields of these prodrugs

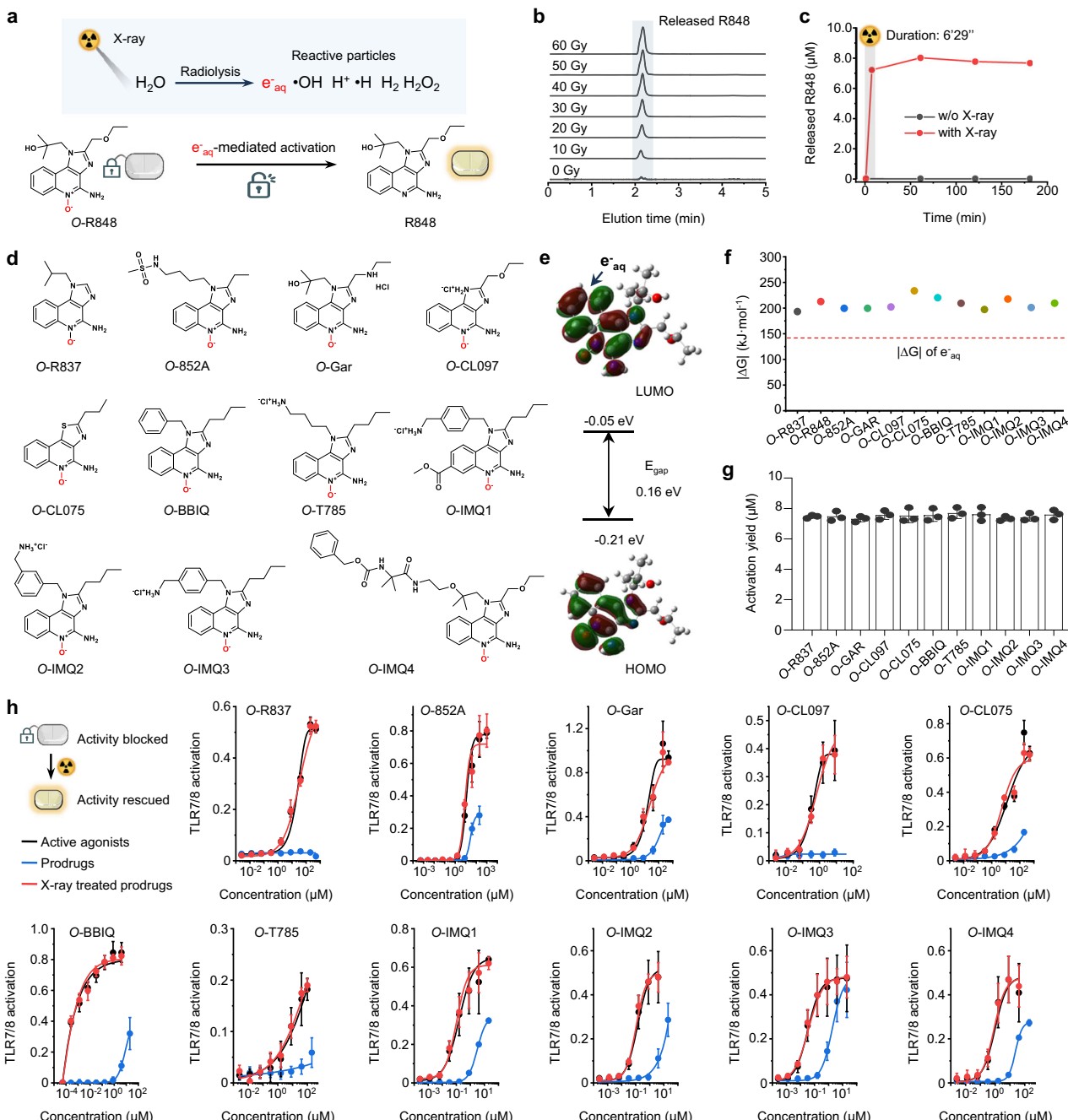

**Fig. 2 | X-ray rescues the activity of oxygen atom-engineered immune agonist prodrugs. a** Schematic illustrations of the $e^-_{aq}$ generated from water radiolysis induces the release of R848 from $O$-R848 through deoxygenation reaction. **b** Radiation dose-dependent release of R848 from $O$-R848 (10 μM in PBS solution) detected by UPLC-MS. **c** Time-dependent release of R848 from $O$-R848 (10 μM in PBS solution) irradiated by 60 Gy X-ray detected by UPLC-MS. **d** Synthesized oxygen atom-engineered immune agonist prodrugs have been assayed for X-ray-induced deoxygenation reaction and immunostimulatory activity rescuing. **e** Molecular orbitals of prodrug ($O$-R848 as a model compound) that indicate $e^-_{aq}$ is

captured by the π-system of imidazoquinoline group). **f** |ΔG| between the anion radical intermediate after capturing $e^-_{aq}$ and the ground state of prodrugs compared with |ΔG| of $e^-_{aq}$. **g** Activation yields of agonists from corresponding prodrugs (10 μM in PBS solution) irradiated by 60 Gy X-ray ($n = 3$ independently tested samples for each group). **h** Concentration-dependent TLR7/8 activation curves of active agonists, and corresponding prodrugs without and with 60 Gy X-ray irradiated. Data were assessed in RAW-Blue report cells after 24 h incubation ($n = 6$ independently tested cell samples for each group), presented as mean values ± s.d. Source data are provided as a Source Data file.

(10 μM in PBS solution) treated with 60 Gy X-ray irradiation were also consistent with the theoretical calculation results (Fig. 2g and Supplementary Figs. 7–17). We next determined whether X-ray can rescue the blocked immunostimulatory activity of these prodrugs. The concentration-dependent TLR7/8 activation of active agonists, corresponding prodrugs, and prodrugs treated with 60 Gy X-ray were compared via the RAW-Blue report cell assay (Fig. 2h). The results

indicated that these reported TLR7/8 agonists hold promise for being developed into new prodrugs for radio-immunotherapy using the SAE-RAP technique.

**Radiotherapy activates $O$-R848 in living cells and living bodies**
Before exploring in vivo tumour treatments of radiotherapy-activated $O$-R848, first it is necessary to validate the immune

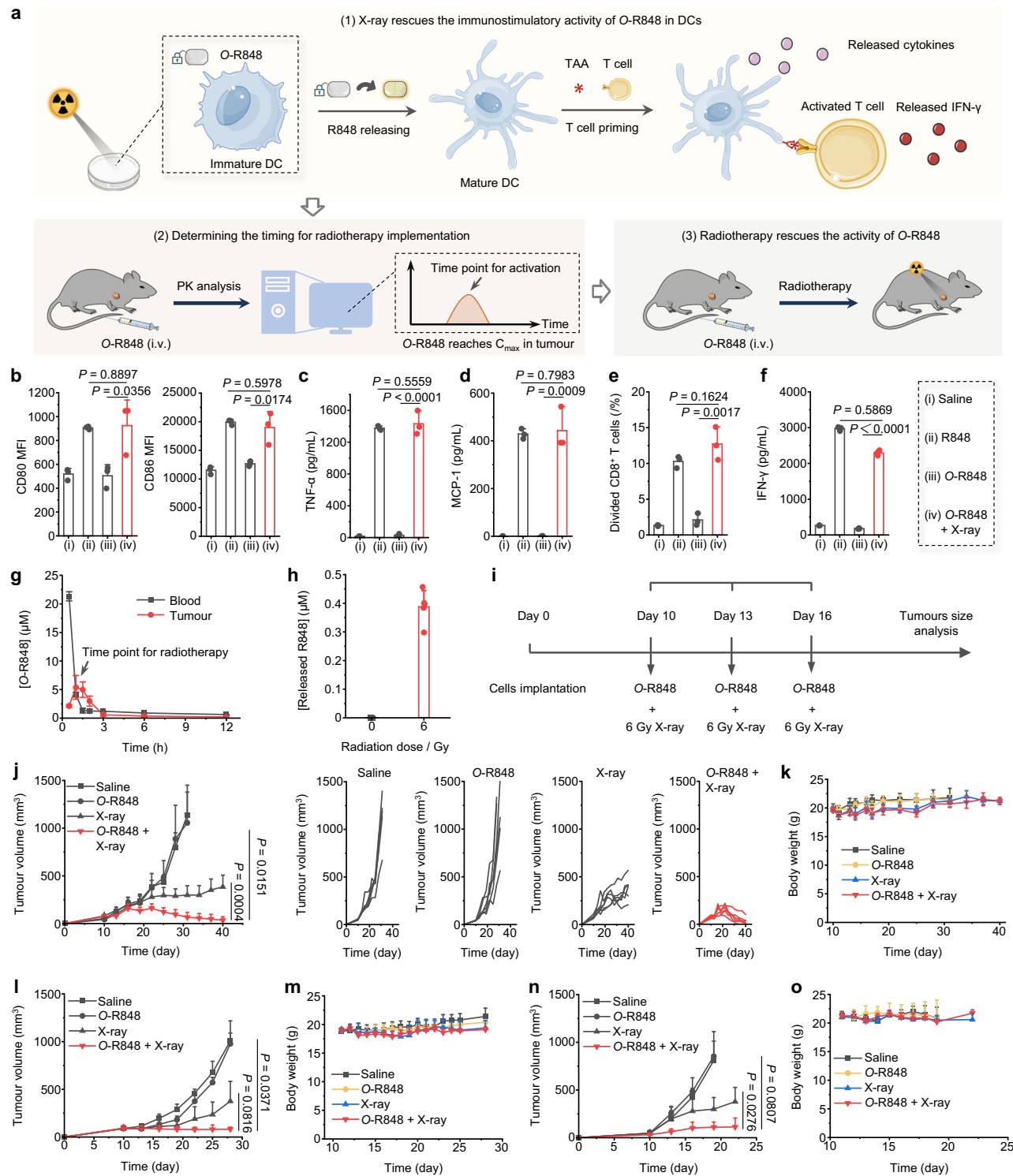

activation mechanism in living antigen-presenting cells (APC). To confirm the generated potency of *O*-R848 after X-ray exposure, we examined the TLR7/8 activation in the RAW-Blue reporter cells. The results indicated that irradiating *O*-R848 together with RAW-Blue cells can activate TLR7/8 equivalently to the positive control R848. However, RAW-Blue cells either incubated alone with *O*-R848 or only exposed to 10 Gy X-ray are insufficient to be effectively activated TLR7/8 (Supplementary Fig. 18a, b). Additionally, we also compared the TLR8 activation of R848, *O*-R848 and *O*-R848 treated with X-ray in human TLR8 reporter HEK293 cells (Supplementary Fig. 19). Furthermore, X-ray treated *O*-R848 could also induce the upregulation

of co-stimulatory markers CD80 and CD86 on RAW264.7 (Supplementary Fig. 18c).

Additionally, dendritic cells (DC) are specialized APCs with a key role in the initiation and regulation of innate and adaptive immune responses. Therefore, we assessed the maturation and functions of bone marrow-derived dendritic cells (BMDC) in response to the activation of *O*-R848 by radiotherapy (Fig. 3a). As a result, co-treatment of bone BMDCs with 1 μM of *O*-R848 and 10 Gy X-ray irradiation significantly increased the expression levels of CD80 and CD86 co-stimulatory markers on BMDCs and enhanced the secretion of TNF and MCP-1 cytokines, thereby facilitating subsequent immune responses

**Fig. 3 | Radiotherapy activates *O*-R848 in living cells and living bodies.**
**a** Schematic illustrations of the experimental design to validate the activation of *O*-R848 by radiotherapy in both living cells and living bodies: (1) In **b**–**f**, *O*-R848 (1 μM) and living BMDCs were irradiated together by 10 Gy X-ray, then the markers expression, released cytokines, and T cells priming were measured after 24 h incubation. Cells incubated with R848 (1 μM) were set as a positive control; (2) Then the time-dependent accumulation of *O*-R848 was analysed post-systemic administration, (3) followed by the implementation of radiotherapy when the *O*-R848 reached its peak concentration ($C_{max}$) in tumours. *O*-R848 was intravenous injected at a dose of 30 μmol/kg and radiotherapy was implemented at a dose of 6 Gy each treatment. **b** CD80/86 expression in BMDCs. Secreted TNF (**c**) and MCP-1 (**d**) from BMDCs were analysed by CBA. **e**, **f** *O*-R848 (1 μM) treated BMDCs were incubated with 10 μg/mL OVA peptides for 24 h, then co-cultured with OT I CD8⁺ T cells for 3 days. Proliferation (**e**) and IFN-γ production (**f**) of OT I CD8⁺ T cells were analysed.

Data in **b**–**f** are presented as mean values ± s.d., *n* = 3 independently tested cell samples for each group, two-tailed unpaired Student's *t*-test. **g** Time-dependent accumulation of *O*-R848 in blood and tumour detected by UPLC-MS (Data are presented as mean values ± s.d., *n* = 3 mice). **h** Concentration of R848 released in tumours treated with or without 6 Gy X-ray irradiation at 1 h after intravenously administrated of *O*-R848 (30 μmol/kg), detected by UPLC-MS (*n* = 5 mice). **i**–**k** Treatment for MC38 bearing C57BL/6J mice. **j** Average tumour growth and tumour volume of an individual mouse. **k** Weight change curves. **l**, **m** Treatment for 4T1 bearing BALB/c mice. **l** Average tumour growth. **m** Weight change curves. **n**, **o** Treatment for B16 bearing C57BL/6 J mice. **n** Average tumour growth. **o** Weight change curves. All mice tested in **j**–**n** were 6–8 weeks old, female, *n* = 6 mice per group. Data are presented as mean values ± s.d., statistical analysis were performed by two-tailed unpaired Student's *t*-test. Source data are provided as a Source Data file.

(Fig. 3b–d). However, 1 μM *O*-R848 alone couldn't sufficiently activate BMDCs, confirming that an oxygen atom effectively blocked the potent immunostimulatory activity of R848. Since DCs play a crucial role in initiating specific T cell-mediated anti-tumour responses to control tumour growth and tumour cell dissemination, we subsequently evaluated the capacity of BMDCs to prime T cells. It is observed that only X-ray-irradiated *O*-R848 could activate BMDCs to enhance T cell proliferation and IFN-γ production (Fig. 3e, f and Supplementary Fig. 20). Collectively, these data confirm that the immunostimulatory activity of *O*-R848 can be rescued by X-ray irradiation to elicit activation of APCs and antigen presentation.

In order to assess the therapeutic efficacy of radiotherapy activating *O*-R848 in vivo, it is crucial to first determine the optimal timing for the implementation of radiotherapy. Theoretically, the timing coincided with the peak concentration ($C_{max}$) of *O*-R848 in tumours could achieve the best therapeutic efficacy (Fig. 3a). As a result, the time-dependent accumulation of *O*-R848 in MC38 tumour tissues indicated that the optimal timing for the implementation of radiotherapy is approximately 1 h after systemic administration (Fig. 3g). Therefore, we selected this timing for subsequent radiotherapy, and at this timing the concentration of R848 released was detected at 387.4 ± 57 nM triggered by 6 Gy X-ray treatment in tumours, which is a sufficient amount for immune activation. Consistent with the human plasma stability results of *O*-R848 (Supplementary Fig. 21), we detected almost no formation of R848 in the blood and tumours during the metabolic time following administration (Fig. 3h and Supplementary Fig. 22). We also tested time-dependent biodistribution of *O*-R848 and formation of R848 in other main tissues (Supplementary Fig. 23), confirming the in vivo stability of *O*-R848. Furthermore, we also assessed the time-dependent distribution of released R848 after radiotherapy (Supplementary Fig. 24). We found R848 exhibits a time-dependent distribution mainly in tumours and also lower blood and kidney distribution for less than 2 h after radiotherapy-activated release from *O*-R848.

Subsequently, we evaluated the activation of *O*-R848 by radiotherapy in tumour-bearing mice. MC38 tumour-bearing mice were randomly divided into four groups, including the control group treated with saline alone, and groups treated with *O*-R848 alone, X-ray alone, and *O*-R848 + X-ray, respectively. The *O*-R848 was intravenously administered into tumour-bearing mice at a dose of 30 μmol/kg and followed with 6 Gy radiotherapy at 1 h post-injection every three days for a total of 3 times (Fig. 3i). After treatments, tumour regression of mice in *O*-R848 + X-ray group was observed (Fig. 3j), as for comparisons, treatments with saline alone, *O*-R848 alone, and X-ray alone could not prevent the growth of tumours. Throughout the treatment process, we continuously monitored body weight and did not observe any significant weight loss (Fig. 3k). Furthermore, we applied this therapeutic strategy to other tumour models, including 4T1 and B16. Intravenous administration of *O*-R848 following activation by radiotherapy, resulted in inhibition of 4T1 tumour growth (Fig. 3l and

Supplementary Fig. 25) without causing significant body weight loss (Fig. 3m). Similarly, in the B16 tumour model, tumour growth was inhibited (Fig. 3n and Supplementary Fig. 26) without significant body weight loss (Fig. 3o). Crucially, radiotherapy activating *O*-R848 in tumours circumvented severe systemic toxicity without causing tissue damage (Supplementary Fig. 27f), and resulted in elongating survival (Supplementary Fig. 27a) in comparison to treatment with R848 directly combined with radiotherapy (Supplementary Fig. 27b–e). These findings indicate that the prodrugs developed through the SAE-RAP technique have the potential to be utilized in reducing systemic toxicity in cancer radio-immunotherapy.

## Radiotherapy activates *O*-R848 in tumours, triggering the "abscopal effect"

It is well established that radiotherapy can induce tumour cell death and promote the production of TAAs. In this study, Radiotherapy also concurrently activates *O*-R848 to release the immunostimulatory of R848, leading to the activation of APCs for enhanced TAA presentation. Consequently, we suppose that radiotherapy activating immune agonist prodrugs can stimulate T cell-mediated anti-immunity to immunologically target unirradiated distant tumours and induce the "abscopal effect", which is rarely seen in preclinical research (Fig. 4a).

To ascertain if the therapeutic effectiveness of radiotherapy-activated *O*-R848 was facilitated by an immune response at the tumour sites, we evaluated the tumour-infiltrating immune cell populations and their activation status using multi-colour flow cytometry. The *O*-R848 was intravenously administered at a dose of 30 μmol/kg and followed by 6 Gy radiotherapy at 1 h post-injection every three days for a total of 3 times. After 3 days, the excised tumours were analysed. The findings showed that radiotherapy-activated *O*-R848 led to an increase in tumour-infiltrating dendritic cells (TIDC), tumour-associated macrophages (TAM), and T cell infiltration, while decreasing the population of immunosuppressive myeloid-derived suppressor cells (MDSC) (Fig. 4b–e). Profiling of the activation states revealed that the immune cells had not only infiltrated the tumour site but were also activated CD86 (Fig. 4b, c), corroborating the in vitro results (Fig. 3b). Further, while no significant differences in Ki67 expression were observed in T cells between X-ray alone group and *O*-R848 + X-ray group (Fig. 4f), radiotherapy-activated *O*-R848 enhanced T cell cytotoxic function, as indicated by increased IFN-γ production (Fig. 4g, h).

In order to investigate whether local radiotherapy could effectively treat abscopal tumours by activating *O*-R848, we established MC38 tumour-bearing mice models with primary and abscopal tumours. Subsequently, 30 μmol/kg of *O*-R848 was intravenously administered and followed with 6 Gy radiotherapy at 1 h post-injection for the primary tumour every three days for a total of 3 times (Fig. 4i). Consistent with previous findings, the tumours treated with the *O*-R848 + X-ray group regressed. Surprisingly, the *O*-R848 + X-ray group exhibited substantial inhibition of growth in abscopal tumours that were not irradiated (Fig. 4j, k). However, X-ray

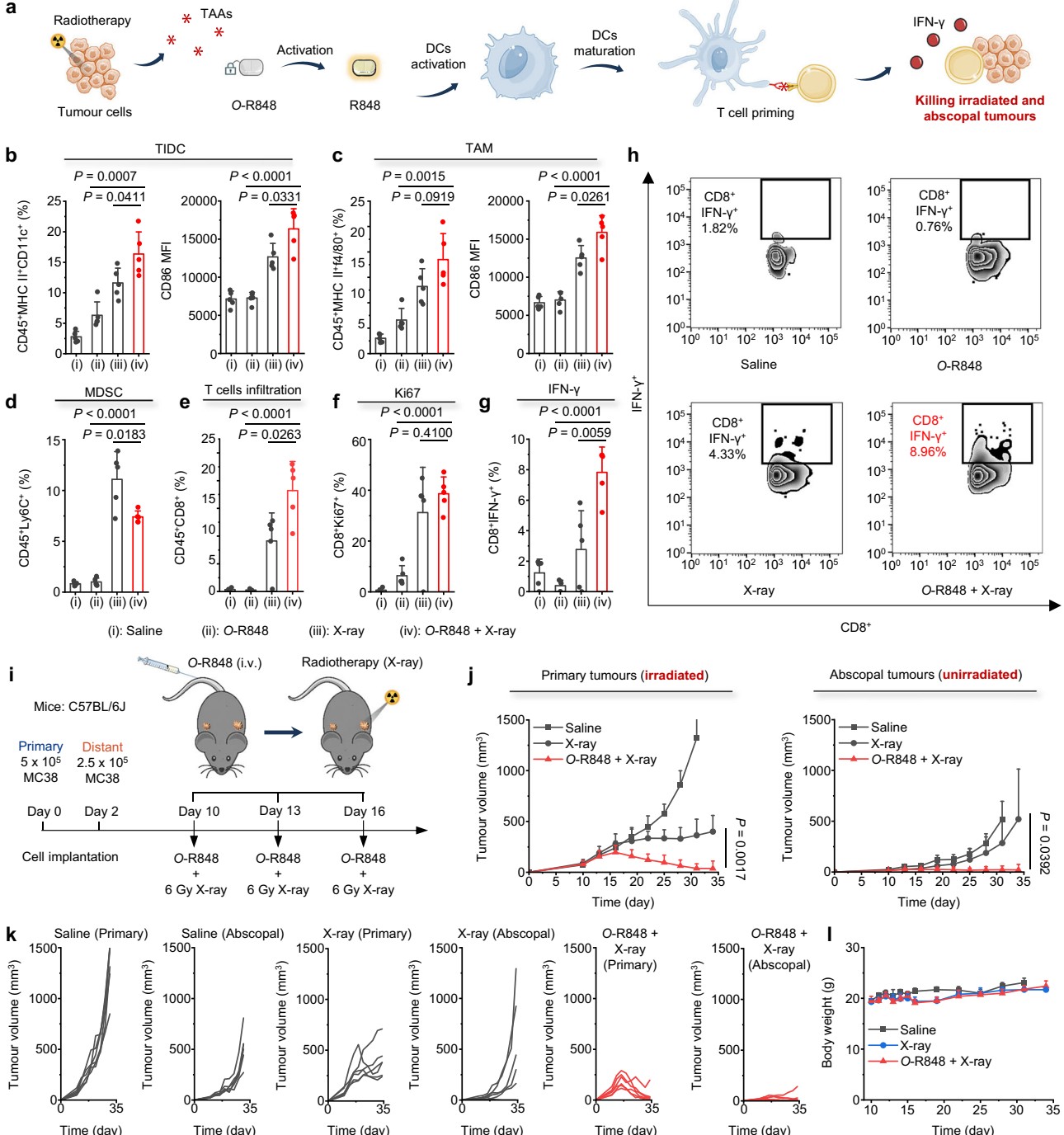

**Fig. 4 | Radiotherapy activates *O*-R848 to elicit anti-tumour immunity, triggering the "abscopal effect".** **a** Schematic illustration of the proposed therapeutic mechanism: Radiotherapy activates *O*-R848, potentiating anti-tumour immunity for treating both irradiated and unirradiated abscopal tumours. **b–h** Immune phenotype analysis by radiotherapy (6 Gy X-ray for each treatment) activated *O*-R848 (30 μmol/kg, by intravenous administration) in MC38 tumour-bearing C57BL/6J mice (6-8 weeks old, female, *n* = 5 mice for each group). **b** Percentages of CD45⁺MHC-II⁺CD11c⁺ TIDCs and expression of CD86. **c** Percentages of CD45⁺MHC-II⁺f4/80⁺ TAMs and expression of CD86. **d–h** Percentages of CD45⁺Ly6C⁺ MDSCs (**d**), CD45⁺CD8⁺ T cells infiltration (**e**), CD8⁺Ki67 (**f**), and CD8⁺IFN-γ (**g**). **h** Representative flow cytometric quantitative results of tumour-infiltrating CD8⁺IFN-γ T cells. **i** Treatment scheme. C57BL/6J mice (6-8 weeks old, female, *n* = 6 mice for each group) were implanted subcutaneously with MC38 cells on day 0 and day 2, followed by intravenous administration of *O*-R848 (30 μmol/kg) and radiotherapy (6 Gy for each treatment). **j** Average volume of irradiated and unirradiated tumours. **k** Tumour volume of an individual mouse. **l** Weight change curves. Data are presented as mean values ± s.d., statistical analysis were performed by two-tailed unpaired Student's *t*-test in (**b–g**, **j**). Source data are provided as a Source Data file.

irradiation alone was inadequate to suppress the growth of unirradiated distant tumours, suggesting that radiotherapy-activated *O*-R848, served as a predominant factor in inhibiting the growth of abscopal tumours. Similarly, radiotherapy activating *O*-R848 did not lead to significant weight loss during the treatment of both primary and abscopal tumours (Fig. 4l).

## SAE-RAP technique as a prodrug developing platform

With the computer-aided SAE-RAP technique, we have developed small-molecule immune agonist prodrugs engineered with a single oxygen atom. This study elucidated the mechanism of activation of oxygen atom-engineered immune agonist prodrugs through a deoxygenation reaction triggered by radiotherapy, and the

therapeutic efficacy of anti-tumour immunity via the model prodrug *O*-R848 activated by radiotherapy both in vitro and in vivo. Therefore, we envision whether the SAE-RAP can also serve as a universal technique for developing radiotherapy-activated prodrugs targeting various receptors.

Firstly, the workflow of computer-aided identifying potential candidates for oxygen atom-engineered prodrugs was designed and shown in Fig. 5a. First, we took advantage of the protein-ligand complex structures in the Protein Data Bank (PDB) bind database and passed them through several filters. For ligands, known drugs (matched with ChEMBL drug dataset) with suitable structures (i.e., nitrogen atom containing six-membered aromatic heterocycles) were selected; for proteins, known targets for cancer treatment (considering compatibility with radiotherapy) were kept. Then the protein-ligand complexes within the intersection were further probed for key contacts by searching interacting residues for each eligible nitrogen atom of the ligand. Finally, 13 representative complexes (including TLR8-R848) were selected on account of the close hydrogen bond distance. We manually installed oxygen atoms onto the key nitrogen atoms and redocked the prodrugs (Fig. 5b). The docking analysis may have resulted in similar poses, because the oxygen atoms were also hydrogen bonding acceptors as the aromatic nitrogen atoms and could thus maintain interactions alike.

To quantify the influence of subtle pose alterations upon the oxygen atom engineering on the structural stability of the 13 complexes, we utilized Molecular mechanics with generalised Born and surface-area solvation (MM-GBSA) methods to calculate the free energy of binding ($\Delta G_{binding}$) as an estimate of affinity. Most of the selected oxygen atom-engineered prodrug molecules exhibited decreased $\Delta G_{binding}$ to various extents (Fig. 5c). Among these representative prodrugs, oxygen atom-engineered camptothecin (*O*-CPT) underwent the greatest $\Delta G_{binding}$ changes, which were theoretically good templates for the SAE-RAP technique. To assess the practical feasibility of computer-aided prodrug design, we conducted experimental investigations to determine the efficacy of the oxygen atom engineering in blocking the activity of CPT, an inhibitor agent of topoisomerase I (TOP I). The cell inhibition rate of CPT and *O*-CPT incubated with MC38, CT26, 4T1, and A549 cell lines, respectively, was measured by CCK-8 assay (Fig. 5d). A single oxygen atom can increase the IC$_{50}$ value of CPT for 11.8 to 22.7-fold in MC38, CT26, 4T1, and A549 cell lines. The results demonstrated that oxygen atom engineering reduces the toxicity of CPT. This finding suggests that small molecules with a CPT-based structure could be formulated as radiotherapy-activated TOP I-targeted prodrugs.

## Discussion

On-target, off-tumour (OTOT) toxicity induced by immunotherapeutic drugs often leads to the failure of clinical trials in the process of drug development. In theory, spatiotemporally controlled activation of drugs in tumour sites has significant potential value, albeit with substantial challenges. Among them, small-molecule immune agonists are potent activators of myeloid cells and lead to the direct killing of tumour cells via enhanced infiltration of cytotoxic T cells. Despite attempts to develop a new type of immune agonists to reduce severe systemic toxicity[32–36], there is still a critical need for an ideal prodrug technique to avoid severe side effects upon systemic exposure and enable tumour-selective, spatiotemporal, and clinically relevant prodrug activation. In this study, with the help of computer-aided drug design, we found that installing a single oxygen atom on the quinoline nitrogen atom can effectively block the immunostimulatory activity of TLR7/8 agonists, reducing the systemic side effects in vivo. Meanwhile, the SAE-RAP technology has also been utilized as a chemical tool to investigate how the radiotherapy-activated immune agonists optimize DC-T cell interactions within the TME to effectively control both local and distal tumours[37].

Based on the finding of the oxygen atom-engineered blockage effect, we demonstrated that the prodrugs can be rescued to regain a high immunostimulatory activity through radiotherapy. Interestingly, radiotherapy is gradually being developed as a precise perturbing tool in life science[38–45]. Therefore, the radiotherapy-activated prodrug (RAP) strategy is promisingly suited to solving the dilemma of immunotherapeutic drug development. Complementarily, local radiotherapy not only kills tumour cells but also activates anti-tumour immunity and facilitates the presentation of tumour-associated antigens (TAAs) generated by radiotherapy, thereby activating T cell immunity. Hence, simple small-molecule prodrugs generated from the Single Atom Engineering for Radiotherapy-Activated Prodrug (SAE-RAP) technique have the potential to resolve current OTOT toxicity challenges in the clinical treatment of radio-immunotherapy, while also providing additional treatments for abscopal tumours. Furthermore, the SAE-RAP technique can also be leveraged to help investigate prodrugs with high atomic efficiency for alternative targets. Here, we used a murine tumour model to evaluate the radiotherapy-activated prodrugs. We acknowledge that the subcutaneous model has limitations in fully assessing efficacy and deep tissue penetration; hence, more appropriate models like orthotopic tumor models are needed in future studies to address these aspects. Future toxicological and pharmacological studies on humanized tumour models would offer more meaningful insights for the clinical translation of the SAE-RAP. There is also a need for further development of molecular engineering technologies to improve the accumulation and retention of prodrugs in tumours.

In summary, our findings demonstrate that the installation of a single oxygen atom on the quinoline nitrogen atom effectively blocks the immunostimulatory activity of TLR7/8 agonists, which can be rescued by radiotherapy to restore high immunostimulatory activity. These results not only provide a promising strategy to overcome severe systemic toxicity in radio-immunotherapy but also present the potential to transform and optimize the current paradigm of combined radiotherapy and immunotherapy.

## Methods

### Reagents

Parent TLR7/8 agonists involved in this work including R848, R837, 852 A, Gardiquimod, CL097, CL075, BBIQ, T785, IMQ1, IMQ2, IMQ3, IMQ4 were purchased from WuXi AppTec in a customized way and used as received. Na$_2$SO$_3$ (AR, #BD151629), NaHCO$_3$ (AR, #BD151377), KNO$_3$ (AR, #P111645), Na$_2$S$_2$O$_3$ (AR, #955588), NaH (AR, #114895), methanol (MeOH) (AR, #BD137672), dichloromethane (DCM) (AR, #BD00959791), CHCl$_3$ (AR, #167735000), hydrochloric acid (HCl, #A04558), anhydrous Na$_2$SO$_4$ (AR, #BD31900), petroleum ether (PE, AR, #HS-003N), NH$_4$OAc (AR, #BD114058), tetrahydrofuran (THF, AR, #962213), ethyl acetate (EtOAc, AR, #937345), di-*tert*-butyl dicarbonate (Boc$_2$O, #BD32840), triethylamine (TEA, #BD155212), 2-(trimethylsilyl)ethoxymethyl chloride (SEM-Cl, BD34874), sodium hydride (NaH, #961241), and *m*-chloroperbenzoic acid (*m*-CPBA, #A84043) were purchased from Energy Chemical, J&K, Innochem, Sigma-Aldrich and used as received. HPLC-grade solvents were purchased from Fisher Scientific (Loughborough, UK), 1x PBS (#M52270-500ML) was purchased from Meryer. Fetal bovine serum (FBS, #SE100-B), penicillin&streptomycin (#15140122), Dulbecco's modified eagle medium (DMEM, #C14190500BT), Roswell Park Memorial Institute-1640 (RPMI-1640, #C11875500BT) medium were purchased from GIBCO and used as received. Cell Counting Kit-8 (CCK-8, #C0040) was purchased from Beyotime Biotechnology Institute. QUANTI-Blue solution (#rep-qbs) was purchased from Invivogen. Ultrapure water was deionized with a Milli-Q SP reagent water system (Millipore) to a specific resistivity of 18.4 MΩ cm. *O*-R837 and *O*-CPT were synthesized according to previously reported literature procedures[24].

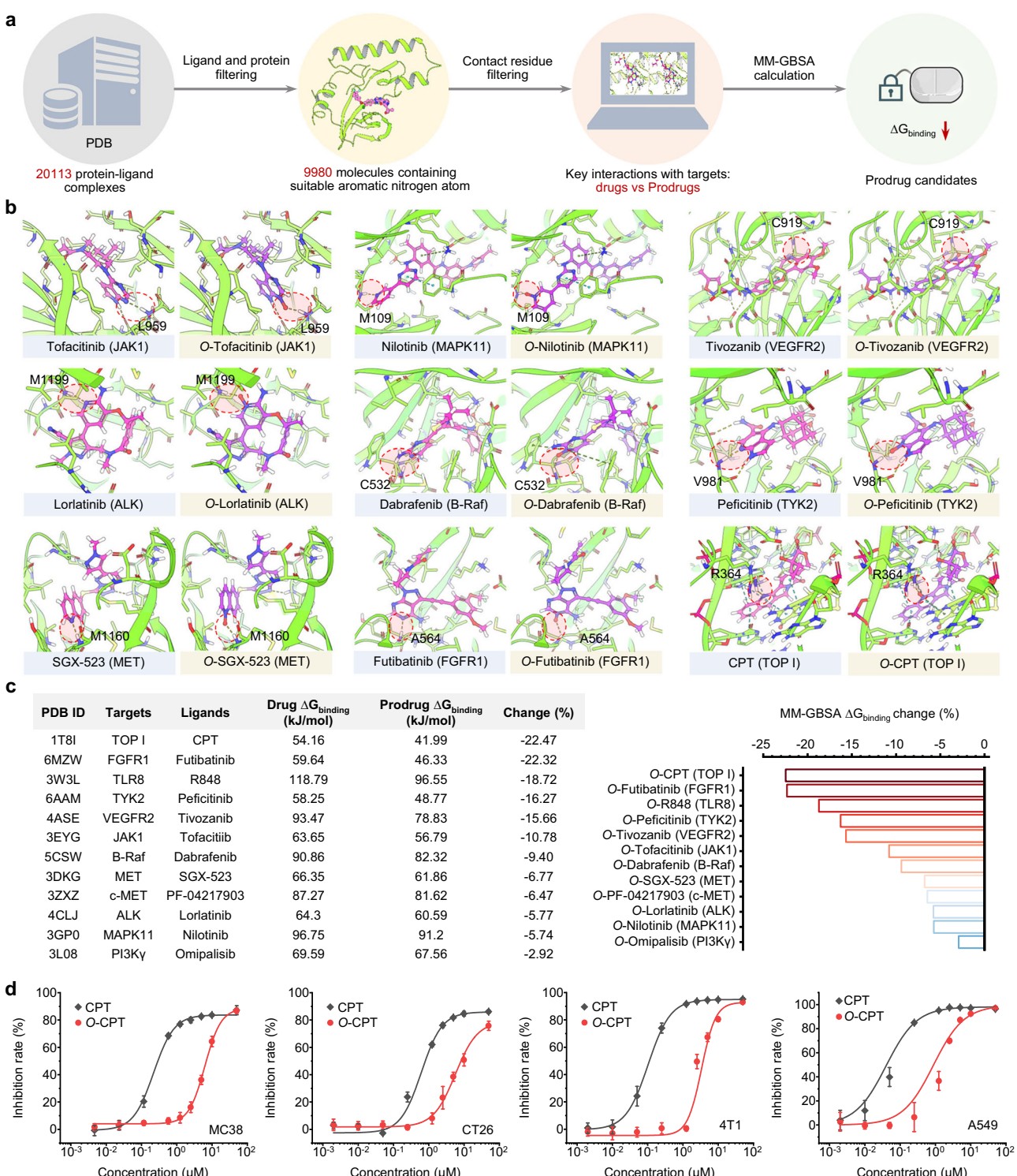

**Fig. 5 | Single-atom-engineering for prodrugs developing platform. a** The workflow of computer-aided identifying potential candidates for oxygen atom-engineered prodrugs. Initially, structural data from protein-ligand complexes in the PDB were employed and subjected to rigorous filtration processes. Then the selection process involved identifying key contacts in protein-ligand complexes by investigating interacting residues around each eligible nitrogen atom in the ligands and oxygen atom-engineered ones. To assess the impact of subtle conformational changes resulting from oxygen atom engineering on the complex's structural stability, the MM-GBSA methods was applied to evaluate the free energy of binding ($\Delta G_{binding}$) as a measure of affinity. **b** Molecular docking analysis of representative parent drugs and oxygen atom-engineered prodrugs against corresponding target proteins, showing the possible disrupting of interaction network by an oxygen atom. **c** $\Delta G_{binding}$ changes between representative drugs and oxygen atom-engineered prodrugs. **d** Cell inhibition rate of CPT and O-CPT incubated with MC38, CT26, 4T1, and A549 cell lines, respectively, measured by CCK-8 assay. $n = 6$ independently tested cell samples for each group. Data are presented as mean values ± s.d. Source data are provided as a Source Data file.

## Apparatus

X-ray irradiation was delivered by an X-ray generator (RS2000 Pro 225, 225 kV, 17.7 mA, Rad Source Technologies, Inc.). Nuclear magnetic resonance (NMR) spectra were recorded on Bruker AVANCE 400 MHz spectrometer. Signals are presented as parts per million (ppm), and multiplicity is presented as single (s), broad (b), doublet (d), triplet (t), quartet (q), or multiplet (m). Ultra-performance liquid chromatography (UPLC-MS) was performed on ACQUITY UPLC H-Class PLUS instrument equipped with Waters PDA eλ Detector and a Waters SQ Detector 2. prep-HPLC (neutral condition, column: Waters Xbridge Prep OBD C18). High-resolution mass spectroscopy was performed on a Bruker Fourier Transform Ion Cyclotron Resonance Mass Spectrometer. Fluorescence spectra were measured on an F-7000 spectrophotometer (Hitachi, Japan). Flow cytometry was performed on BD FACSLyric™ Clinical. The absorbance of cells was measured on TECAN Infinite E Plex. The residue analysis, molecule docking, and interaction analysis were performed with Molecular Operating Environment (MOE, Canada), GOLD (CCDC, UK) and Discovery Studio Visualizer (BIOVIA), respectively. The molecular dynamics (MD) was performed with Assisted Model Building with Energy Refinement (AMBER 20).

## General synthetic methods of oxygen-engineered agonists *O*-R848, *O*-852A, *O*-CL075, *O*-BBIQ, *O*-IMQ4

To a solution of R848 (100.0 mg, 318.1 μmol, 1.0 eq.) in DCM (2 mL) was added *m*-CPBA (96.9 mg, 477.1 μmol, 85% purity, 1.5 eq.) at 0 °C. The mixture was stirred at room temperature for 12 h, and then added to saturated sodium thiosulfate aqueous solution (1 mL), concentrated under reduced pressure to give a residue. The residue was purified by prep-HPLC to give *O*-R848 (54.4% yield) as a light-yellow solid. $^1$H NMR (400 MHz, DMSO-$d_6$): δ = 8.58-8.51 (m, 1H), 8.45 (d, $J$ = 8.3 Hz, 1H), 7.68-7.48 (m, 3H), 7.44 (ddd, $J$ = 8.3, 7.0, 1.4 Hz, 1H), 4.90 (s, 1H), 4.70 (s, 2H), 3.58 – 3.34 (m, 4H) 1.14 (q, $J$ = 6.9 Hz, 9H). $^{13}$C NMR (101 MHz, DMSO): δ = 153.14, 143.37, 137.39, 128.13, 126.21, 123.71, 122.73, 118.64, 113.26, 71.18, 65.96, 65.21, 55.50, 28.11, 15.50. HRMS (ESI +): calculated for $C_{17}H_{23}N_4O_3^+$ ([M + H]$^+$): 331.1764, found: 331.1765.

## *O*-852A, *O*-CL075, *O*-BBIQ, and *O*-IMQ4 were synthesized according to similar procedures

*O*-852A (56.3% yield) was obtained as a brown solid. $^1$H NMR (400 MHz, Methanol-$d_4$): δ = 8.52 (d, $J$ = 8.6 Hz, 1H), 8.27 (d, $J$ = 8.2 Hz, 1H), 7.75 (t, $J$ = 7.8 Hz, 1H), 7.62 (t, $J$ = 7.7 Hz, 1H), 4.64 (d, $J$ = 7.9 Hz, 2H), 3.15 (t, $J$ = 6.6 Hz, 2H), 3.06 (q, $J$ = 7.4 Hz, 2H), 2.91 (s, 3H), 2.07 – 1.97 (m, 2H), 1.77 (p, $J$ = 6.9 Hz, 2H), 1.51 (t, $J$ = 7.4 Hz, 3H). $^{13}$C NMR (101 MHz, MeOH): δ = 156.99, 144.25, 136.29, 128.88, 128.36, 124.80, 120.83, 117.35, 112.49, 44.96, 41.88, 38.11, 26.52, 20.14, 10.48. HRMS (ESI +): calculated for $C_{17}H_{24}N_5O_3S^+$ ([M + H]$^+$): 378.1594, found: 378.1594.

*O*-CL075 (58.4% yield) was obtained as a yellow solid. $^1$H NMR (400 MHz, DMSO-$d_6$): δ = 8.44 (d, $J$ = 8.5 Hz, 1H), 8.02-7.95 (m, 1H), 7.83-7.66 (m, 3H), 7.50 (t, $J$ = 7.5 Hz, 1H), 3.18 (t, $J$ = 7.4 Hz, 2H), 1.88 (h, $J$ = 7.5 Hz, 2H), 1.02 (t, $J$ = 7.3 Hz, 3H). $^{13}$C NMR (101 MHz, MeOH): δ = 174.07, 144.98, 136.42, 134.81, 130.18, 125.15, 117.04, 116.69, 35.37, 22.76, 12.51. HRMS (ESI +): calculated for $C_{13}H_{14}N_3OS^+$ ([M + H]$^+$): 260.0852, found: 260.0852.

*O*-BBIQ (51.3% yield) was obtained as a light-yellow solid. $^1$H NMR (400 MHz, DMSO-$d_6$): δ = 8.49 (d, $J$ = 8.6 Hz, 1H), 7.91 (d, $J$ = 8.3 Hz, 1H), 7.55 (t, $J$ = 7.9 Hz, 3H), 7.29 (dt, $J$ = 20.2, 7.4 Hz, 4H), 7.03 (d, $J$ = 7.5 Hz, 2H), 5.91 (s, 2H), 2.93 (t, $J$ = 7.7 Hz, 2H), 1.70 (p, $J$ = 7.6 Hz, 2H), 1.36 (h, $J$ = 7.3 Hz, 2H), 0.84 (t, $J$ = 7.4 Hz, 3H). $^{13}$C NMR (101 MHz, DMSO): δ = 156.23, 143.44, 137.08, 136.84, 129.61, 128.16, 127.94, 126.84, 126.64, 126.14, 124.19, 121.34, 118.86, 112.48, 48.81, 30.13, 26.87, 22.42, 14.28. HRMS (ESI +): calculated for $C_{21}H_{23}N_4O^+$ ([M + H]$^+$): 347.1866, found: 347.1866.

*O*-IMQ4 (45.4% yield) was obtained as a white solid. $^1$H NMR (400 MHz, DMSO-$d_6$): δ = 8.54 (d, $J$ = 8.6 Hz, 1H), 8.48 (d, $J$ = 8.3 Hz, 1H), 7.67-7.53 (m, 3H), 7.46 (t, $J$ = 7.6 Hz, 1H), 7.37 – 7.28 (m, 5H), 7.28 (s, 1H), 7.04 (d, $J$ = 6.4 Hz, 1H), 4.95 (s, 2H), 3.52 (t, $J$ = 7.0 Hz, 2H), 3.36 (s, 4H), 3.12 (s, 2H), 2.87 (d, $J$ = 7.0 Hz, 2H), 1.20 (s, 6H), 1.13 (t, $J$ = 7.0 Hz, 9H). $^{13}$C NMR (101 MHz, DMSO): δ = 174.51, 155.06, 153.18, 143.36, 137.41, 128.80, 128.21, 128.07, 126.29, 123.55, 123.05, 118.73, 113.36, 76.49, 66.00, 65.62, 65.16, 60.19, 56.31, 55.24, 25.65, 22.50, 15.51. HRMS (ESI +): calculated for $C_{31}H_{41}N_6O_6^+$ ([M + H]$^+$):593.3083, found:593.3082.

## General synthetic methods of oxygen-engineered agonists *O*-T785, *O*-Gar, *O*-IMQ1, *O*-IMQ2, *O*-IMQ3

To a solution of T785 (1.0 g, 3.2 mmol, 1.0 eq.) in 20 mL CHCl$_3$ was added a solution of Boc$_2$O (0.13 g, 0.6 mmol, 0.2 eq.) in 10 mL of CHCl$_3$ over 4 h. After stirring for an additional 14 h, the resulting slurry was diluted with DCM and washed with 1 N NaHCO$_3$. The organic layer was dried over Na$_2$SO$_4$, filtered, and concentrated to give C1 (531.0 mg) as a yellow solid.

To a mixture of C1 (500.0 mg, 1.2 mmol, 1.0 eq.) in DCM (6 mL) was added *m*-CPBA (246.7 mg, 1.2 mmol, 85% purity, 1.0 eq.) at 0 °C, then the mixture was stirred at 0 °C for 0.5 h. The reaction mixture was quenched by sat. Na$_2$S$_2$O$_3$ aq. (10 mL), extracted with EtOAc (50 mL × 3). The combined organic layers were washed with 50 mL of sat. NaHCO$_3$ aq., dried over anhydrous Na$_2$SO$_4$, filtered, and concentrated under reduced pressure to give a residue. The residue was purified by column chromatography on silica gel to give C2 (176.0 mg, crude) as black oil.

A mixture of C2 (300.0 mg, 701.7 μmol, 1.0 eq.) in EtOAc/HCl (2 mL) was stirred at 25 °C for 0.5 h. The reaction mixture was concentrated to give the crude product, then purified by prep-HPLC to give *O*-T785 (42.5% yield) as a yellow solid. $^1$H NMR (400 MHz, DMSO-$d_6$): δ = 9.38 (s, 1H), 9.14 (s, 1H), 8.26 (dd, $J$ = 8.3, 1.3 Hz, 1H), 8.17 (s, 2H), 8.09 (dd, $J$ = 8.6, 1.2 Hz, 1H), 7.79 (ddd, $J$ = 8.4, 7.2, 1.1 Hz, 1H), 7.64 (ddd, $J$ = 8.4, 7.2, 1.2 Hz, 1H), 4.61 (t, $J$ = 7.3 Hz, 2H), 2.99 (dd, $J$ = 8.5, 7.1 Hz, 2H), 2.80 (hept, $J$ = 6.1, 5.2 Hz, 2H), 1.93-1.68 (m, 6H), 1.46 (h, $J$ = 7.4 Hz, 2H), 0.95 (td, $J$ = 7.4, 1.6 Hz, 3H). $^{13}$C NMR (101 MHz, DMSO): δ = 157.20, 147.64, 135.04, 132.34, 129.94, 126.11, 124.63, 122.19, 116.42, 112.24, 45.40, 38.72, 29.78, 27.13, 26.67, 24.45, 22.37, 14.30. HRMS (ESI +): calculated for $C_{18}H_{26}N_5O^+$ ([M + H]$^+$):328.2132, found:328.2132.

## *O*-Gar, *O*-IMQ1, *O*-IMQ2, and *O*-IMQ3 were synthesized according to similar procedures

*O*-Gar (41.4% yield) as a brown solid. $^1$H NMR (400 MHz, Methanol-$d_4$): δ = 8.55 (d, $J$ = 8.3 Hz, 1H), 8.19 (d, $J$ = 8.6 Hz, 1H), 7.87 (t, $J$ = 7.9 Hz, 1H), 7.69 (t, $J$ = 7.7 Hz, 1H), 4.84 (s, 4H), 3.38 (q, $J$ = 7.3 Hz, 2H), 1.46 (t, $J$ = 7.3 Hz, 3H), 1.32 (s, 6H). $^{13}$C NMR (101 MHz, MeOH): δ = 150.17, 148.08, 135.27, 134.51, 130.38, 125.58, 124.46, 122.72, 115.26, 112.76, 71.08, 55.57, 43.14, 26.31, 10.25. HRMS (ESI +): calculated for $C_{17}H_{24}N_5O_2^+$ ([M + H]$^+$):330.1924, found:330.1925.

*O*-IMQ1 (29.3% yield) as a white solid. $^1$H NMR (400 MHz, Methanol-$d_4$): δ = 8.70 (d, $J$ = 1.6 Hz, 1H), 8.06 (d, $J$ = 8.6 Hz, 1H), 7.95 (dd, $J$ = 8.6, 1.7 Hz, 1H), 7.48 (d, $J$ = 8.0 Hz, 2H), 7.20 (d, $J$ = 7.9 Hz, 2H), 6.03 (s, 2H), 4.09 (s, 2H), 3.97 (s, 3H), 3.03 (t, $J$ = 7.6 Hz, 2H), 1.87 (p, $J$ = 7.6 Hz, 2H), 1.48 (h, $J$ = 7.4 Hz, 2H), 0.95 (t, $J$ = 7.4 Hz, 3H). $^{13}$C NMR (101 MHz, MeOH): δ = 165.40, 158.72, 148.54, 135.97, 134.59, 133.23, 132.48, 130.67, 129.72, 126.13, 125.25, 122.10, 116.29, 115.12, 51.91, 42.37, 28.99, 26.44, 21.92, 12.66. HRMS (ESI +): calculated for $C_{24}H_{28}N_5O_3^+$ ([M + H]$^+$):434.2186, found:434.2187.

*O*-IMQ2 (58.4% yield) as a white solid. $^1$H NMR (400 MHz, DMSO-$d_6$) δ 9.47 (s, 1H), 9.29 – 9.14 (m, 1H), 8.52 (s, 2H), 8.05 (dd, $J$ = 8.7, 1.2 Hz, 1H), 7.95 (dd, $J$ = 8.3, 1.2 Hz, 1H), 7.69 (ddd, $J$ = 8.6, 7.2, 1.2 Hz, 1H), 7.47 – 7.36 (m, 3H), 7.24 (d, $J$ = 1.9 Hz, 1H), 7.19 – 7.10 (m, 1H), 5.94 (s, 2H), 3.92 (q, $J$ = 5.6 Hz, 2H), 2.97 (t, $J$ = 7.7 Hz, 2H), 1.82 – 1.70 (m, 2H), 1.40 (h, $J$ = 7.4 Hz, 2H), 0.88 (t, $J$ = 7.3 Hz, 3H). $^{13}$C NMR (101 MHz, DMSO): δ = 157.71, 147.78, 136.36, 136.55, 135.18, 132.96, 129.96, 129.83, 128.94, 126.62, 126.27, 125.79, 125.00, 122.15, 116.34, 112.13, 48.91, 42.33, 29.72, 26.75, 22.30, 14.24. HRMS (ESI +): calculated for $C_{22}H_{26}N_5O^+$ ([M + H]$^+$):376.2130, found:376.2132.

*O*- IMQ3 (62.9% yield) as a white solid. $^1$H NMR (400 MHz, DMSO-$d_6$) δ 9.31 (d, $J$ = 104.6 Hz, 2H), 8.47 (s, 2H), 8.05 (d, $J$ = 8.6 Hz, 1H), 7.97 (d, $J$ = 8.3 Hz, 1H), 7.67 (t, $J$ = 7.9 Hz, 1H), 7.46 (d, $J$ = 8.1 Hz, 2H), 7.39 (t, $J$ = 7.7 Hz, 1H), 7.10 (d, $J$ = 8.0 Hz, 2H), 5.98 (s, 2H), 3.94 (d, $J$ = 5.0 Hz, 2H), 2.96 (t, $J$ = 7.7 Hz, 2H), 1.73 (p, $J$ = 7.6 Hz, 2H), 1.38 (h, $J$ = 7.4 Hz, 2H), 0.87 (t, $J$ = 7.3 Hz, 3H). $^{13}$C NMR (101 MHz, DMSO): δ = 157.56, 147.67, 136.15, 135.16, 134.03, 132.68, 130.13, 129.89, 126.22, 125.54, 125.01, 122.02, 116.30, 112.02, 48.65, 42.11, 29.78, 26.68, 22.21, 14.14. HRMS (ESI + ): calculated for $C_{22}H_{26}N_5O^+$ ([M + H]$^+$):376.2132, found:376.2132.

## Synthetic method of oxygen-engineered agonist *O*-CL097

To an ice-cold, argon-flushed solution of the respective CL097 in dry THF (1.0 g, 4.1 mmol, 1.0 eq.), NaH (0.18 g, 60% dispersion in mineral oil, 1.1 eq.) was added portion wise and stirred at 0 °C for 30 min. SEM-Cl (0.89 g, 5.4 mmol in toluene, 1.3 eq.) was added dropwise, and the solution was stirred at room temperature for 30 min. The reaction was quenched with saturated $NH_4OAc$ solution and stirred for a further 30 min. The mixture was extracted three times with EtOAc, dried over anhydrous $Na_2SO_4$ and the solvent was removed in vacuo. The crude product was purified via flash chromatography to obtain C3.

Then to a mixture of C3 (2.5 g, 6.7 mmol, 1.0 eq.) in DCM (30 mL) was added *m*-CPBA (1.36 g, 6.7 mmol, 85% purity, 1.0 eq.) at 0 °C, then the mixture was stirred at 0 °C for 30 min. The reaction mixture was quenched by sat. $Na_2S_2O_3$ aq. (50 mL), extracted with EtOAc (150 mL × 3). The combined organic layers were washed with 200 mL of sat. $NaHCO_3$ aq., dried over anhydrous $Na_2SO_4$, filtered and concentrated under reduced pressure to give a residue, and purified by column chromatography on silica gel to give C4 (1.8 g, crude) as yellow oil.

A mixture of C4 (500.0 mg, 1.3 mmol, 1.0 eq.) in EtOAc/HCl (5 mL) at 25 °C, then the mixture was stirred at 25 °C for 2 h. The mixture was then filtered, and the filter cake was dried in a vacuum to give *O*-CL097 (57.2% yield) as a white solid. $^1$H NMR (400 MHz, Methanol-$d_4$): δ = 8.28 (dd, $J$ = 8.1, 1.4 Hz, 1H), 8.11 (d, $J$ = 8.7 Hz, 1H), 7.84 (ddd, $J$ = 8.7, 7.3, 1.4 Hz, 1H), 7.66 (td, $J$ = 7.7, 7.3, 1.1 Hz, 1H), 4.83 (s, 2H), 3.72 (q, $J$ = 7.0 Hz, 2H), 1.31 (t, $J$ = 7.0 Hz, 3H). $^{13}$C NMR (101 MHz, MeOH): δ = 153.77, 148.04, 135.08, 134.92, 130.50, 125.80, 122.87, 122.38, 114.88, 112.24, 66.77, 65.09, 14.02. HRMS (ESI + ): calculated for $C_{13}H_{15}N_4O_2^+$ ([M + H]$^+$):259.1192, found:259.1190.

## Computational methods

All the density functional theory (DFT) calculations were performed with the Gaussian 09 series of programs. The B3LYP functional with the 6-311 + G(d,p) basis set was used to calculate the single-point energies in the water solvent to provide more accurate energy information. The solvent effect was considered by single-point calculations based on the gas-phase oxygen-engineered agonists with the SMD continuum solvation model. The Gibbs free energies of the oxygen-engineered agonists calculated using the B3LYP functional are used to discuss the energies. The molecular orbital energies and distributions of the oxygen-engineered agonists, especially HOMO and LUMO, calculated using the B3LYP functional are used to study.

## Calculation on comparing the reactivity of oxygen-engineered agonists with hydrated electrons

It has been known that the reaction of hydrated electrons with oxygen-engineered agonists is diffusion-controlled, it is reliable to consider only thermodynamics. The electron affinity of oxygen-engineered agonists (ΔG) refers to the free energy change of oxygen-engineered agonist to its anion (1 Hartree = 2625.5 kJ/mol):

$$\Delta G = [O]^- - O \qquad (1)$$

Comparing the ΔG with the enthalpy of the formation of hydrated electrons, the reactivity of oxygen-engineered agonists can be judged. Electron affinity can be obtained by calculating the Gibbs free energies

of oxygen-engineered agonists and their anions, respectively. Comparing ΔG with the absolute free energy of solvation of electrons (ΔG$_e$ = 35.5 kcal/mol) in water, it is possible to judge whether the reaction can occur. Using electrode potentials (φ) to represent the reactivity of different oxygen-engineered agonists:

$$\varphi = -\frac{\Delta G - \Delta G_e}{F} \qquad (2)$$

## Molecular docking and kinetic simulation

Informatics analyses of the PDB bind datasets were done using RDKit and Biopython. Molecular modelling tasks were performed using various modules in Schrödinger Suites 2018-1 (Schrödinger, LLC, NY). The co-crystal structure of R848 with the activated state of human TLR8 dimers (PDB ID: 3W3L) was retrieved from the RCSB Protein Data Bank and was then prepared and minimized using the Protein Preparation Wizard. An oxygen atom was directly added onto the crystal structure of R848 to form *O*-R848, which was subsequently docked into the binding site of R848 using Glide. The refined ternary structure of TLR8 dimer-R848 and the docked structure of TLR8 dimer-*O*-R848 were then solvated and neutralized. Molecular dynamics simulations were performed in Desmond. Each of the two initial models was first relaxed using a six-step default protocol and was then subjected to a 30 ns NPT simulation with temperature fixed at 300 K and pressure at 1.01 bar. After each simulation was completed, RMSD fluctuations and protein-ligand interactions could be calculated in the Simulation Interaction Diagram module based on the resulting trajectories. For protein-ligand complexes, MM-GBSA calculations were performed in Prime, where the VSGB solvation model was selected and residues within 3 Å from the ligand were treated as flexible.

## Radiation delivery and in vivo radiotherapy

For the in vitro test, degassed test tubes were X-ray-irradiated with total doses of 0-60 Gy at the rate of 6.98 Gy/min. Living cells were degassed and X-ray-irradiated with total doses of 10 Gy at the rate of 2.87 Gy/min. Local irradiation of the implanted tumour was conducted using a customized mouse jig. Tumours were locally irradiated at a dose of 6 Gy at a dose rate of 2.87 Gy/min with other parts of the body shielded with 5 mm thick lead.

## Cells culture

The MC38 cell lines (#SNL-505) were purchased from SUNNCELL. The B16 (#CL-0029), 4T1 (#CL-0007), A549 (#CL-0016), CT26 (#CL-0071), HEK 293 (#CL-0001) and RAW264.7 (#CL-0190) cell lines were purchased from Procell. RAW-Blue reporter cells (RAW-Blue™ Cells) were purchased from Invivogen. QUANTI-Blue solution (#rep-qbs) was purchased from Invivogen. MC38, B16, A549, HEK 293 reporter cells, RAW 264.7, and RAW-Blue reporter cells were cultured in DMEM medium. 4T1 and CT26 cells were cultured in RPMI-1640 medium. The media were supplemented with 10% (v/v) fetal bovine serum (FBS), penicillin (100 units/mL), and streptomycin (100 μg/mL). All these cells were cultured in a 5% $CO_2$ incubator at 37°C and the medium was replaced every 2-3 days. After growing to 80% confluence, the cells were treated with trypsin or cell scraper and then seeded on dishes, 96-well, or 6-well plates overnight for further experiments.

## RAW-Blue or HEK 293 reporter cell line assays

RAW-Blue or HEK 293 reporter cells were plated in a 96-well plate at a density of 5 × 10$^4$ cells/well. The cells were treated with a series of concentrations of drug molecules for 24 h. Then, 20 μL of supernatant from each well was extracted and mixed with 180 μL of QUANTI-Blue solution. The absorbance was measured at 630 nm via a microplate reader.

## Cell viability assays

Cell viability was assessed with the CCK-8 assay following the protocol, and treatment groups were normalized to controls. To assay the cytotoxicity of $O$-R848 and R848, MC38 and RAW 264.7 cells were seeded in a 96-well plate at a concentration of $1 \times 10^4$/mL in 100 μL of DMEM medium with 10% FBS and 1% penicillin/streptomycin and maintained at 37 °C for 24 h. Then the cells were incubated with different concentrations of $O$-R848 and R848 for 24 h. Then the medium of each well was replaced by a blank medium containing a final concentration of 0.5 mg/mL CCK-8. The cells were incubated at 37 °C for 2 h, and the absorbance was measured at 450 nm. The absorbance of treated cells was compared with the absorbance of the control group, of which the viability was set as 100%. To assay the cytotoxicity of $O$-CPT and CPT, MC38, 4T1 and A549 cells were seeded in a 96-well plate at a concentration of $1 \times 10^4$/mL in 100 μL of DMEM medium (except for 4T1 and CT26 in RPMI-1640 medium) with 10% FBS and 1% penicillin/streptomycin and maintained at 37 °C for 24 h. Then the cells were incubated with different concentrations of $O$-CPT and CPT for 72 h. Then the medium of each well was replaced by a blank medium containing a final concentration of 0.5 mg/mL CCK-8. The cells were incubated at 37 °C for 2 h and the absorbance was measured at 450 nm. The absorbance of treated cells was compared with the absorbance of the control group, of which the viability was set as 100%.

## Antibody information

Anti-mouse CD8α-APC ($\leq 0.125$ μg per million cells in 100 μL volume, Biolegend, #126614, clone YTS156.7.7), anti-mouse CD3-BV421($\leq 0.25$ μg per million cells in 100 μL volume, Biolegend, #100228, clone 17A2), anti-CD16/32 antibody($\leq 1.0$ μg per million cells in 100 μL volume, Biolegend, #101301, clone 93), APC anti-mouse IFN-γ Antibody($\leq 1.0$ μg per million cells in 100 μL volume, BioLegend, #505809, clone XMG1.2), anti-mouse Ki67($\leq 0.5$ μg per million cells in 100 μL volume, Biolegend, #652401, clone 16A8), The antibodies involved in the experiment have been organized and listed in Supplementary Table 3.

## BMDCs generation and activation

BMDCs were generated by culturing bone-marrow cells in the presence of rmGM-CSF (20 ng/ml, Biolegend, #576306) for 6 days. Then the BMDCs were treated with saline, R848 alone, $O$-R848 alone, and $O$-R848 + X-ray (10 Gy). After 24 h incubation, cells were sorted, and surface markers CD80/86 were analysed on flow cytometry (BD Bioscience, #561135, #561964). The supernatant was collected and measured by CBA (BD, #552364).

## T cells priming assay

OT I CD8$^+$ T cells were isolated from spleens using magnetic beads (Stem cells). T cells were loaded with 2.5 uM CFSE (BD, #565082), then cocultured with antigen and X-ray-activated agonist prodrugs treated DCs at 8:1. After 72 h incubation, T cells were analysed after staining with anti-mouse CD8α-APC and anti-mouse CD3-BV421 by flow cytometry to assess CFSE dilution. The supernatant was collected and IFN-γ was measured by CBA.

## Animal ethics

All animal care and experimental procedures were performed by following the animal protocols (IACUC ID: CCME-LiuZB-2) approved by the ethics committee of Peking University. Ethical compliance with the IACUC protocol was maintained. 6-8-week-old female (Considering that female mice exhibit lower aggression when living in groups, which are more suitable for group breeding with reduced management difficulty) C57BL/6 J (#213) and BALB/c (#211) mice were purchased from Beijing Vital River Laboratory Animal Technology Co. Ltd., and maintained under specific-pathogen-free facility (SPF) conditions with a 12 light/12 dark cycle, and free access to food and water. Mice were housed under a temperature of 24 ± 2 °C, and humidity of 50 ± 10%. The

experimental and control animals were co-housed. In none of the experiments did the size of the tumour graft surpass 2 cm in any two dimensions (according to the limits defined by the IACUC protocol). For euthanasia, the mice were euthanized by carbon dioxide according to the approved protocols.

## General protocols of animal studies

The exact sample size is stated in the figure legend. In animal experiments to detect tumour growth, for individual analyses $n = 6$ mice in each group. For all other experiments, for individual analyses $n = 3, 5$, or 6 mice in each group, were selected on the basis that the variability between estimates is sufficiently small to provide significant differences between test samples in these studies). No data were excluded from the analyses. All results were repeated at least three times independently with similar results. Mice were randomly assigned to experimental groups (the randomization was applied following these two criteria: the probability of assignment to any of the experimental groups is equal for each subject and the assignment of one subject to a group does not affect the assignment of any other subject to that same group). Proper blinding was applied during the data collection and analysis.

## Tumour implantation and treatments

MC38 and B16 cells ($5 \times 10^5$) were injected subcutaneously into the right flank of 6-8 weeks old C57BL/6 J mice. 4T1 cells ($5 \times 10^5$) were injected subcutaneously into the right flank of 6-8-week-old BALB/c mice. After the tumour was established, mice were randomized into treatment groups ($n = 6$) with saline or 30 μmol/kg $O$-R848 (intravenous injection) every three days and/or radiation (6 Gy, 3 times in total). For the abscopal model, MC38 ($5 \times 10^5$) cells were injected subcutaneously into the right flank. 2 days later, MC38 ($2.5 \times 10^5$) cells were injected subcutaneously into the left flank, and then intravenous injection of 30 μmol/kg $O$-R848 and radiation only focused on the primary (right) tumour. Tumour volumes were measured twice a week and calculated by length × width × height/2.

## Determination of time-dependent accumulation of $O$-R848 and R848 in vivo

MC38 tumour-bearing mice ($n = 5$) were intravenously injected with $O$-R848 (30 μmol/kg). The mice were treated with or without radiotherapy (6 Gy), then sacrificed at pre-set time points post-injection and tumour tissues were collected, and wet weighed. Tumours were ground in a microwave digestion system, 100 μL slurry or blood was exacted with 750 μL MeOH and 250 uL MeCN. The concentrations of $O$-R848 and R848 were measured by UPLC-MS.

## Immune infiltration in tumours analysed by flow cytometry

Tumours were excised and mechanically dissociated into single-cell suspensions using type IV collagenase (2 mg/ml, Gibco, #17104019) and deoxyribonuclease I (100 μg/ml, Invitrogen, #18047019). Cells were resuspended in FACS buffer (1% bovine serum albumin and 0.05% NaN$_3$) were blocked with anti-CD16/32 antibody for 15 min and then stained with specific surface antibodies for 30 min at 4 °C. For intracellular IFN-γ and Ki67 staining, samples were resuspended with Fixation/Permeabilization buffer (eBioscience, #88-8824-00), incubated for 60 min at room temperature or overnight at 4 °C in the dark, then washed twice with permeabilization buffer, and stained with anti-mouse IFN-γ, anti-mouse Ki67, or LIVE/DEAD fixable yellow dye (Thermo Fisher Scientific, #L34959) for 30 min. Samples were analyzed on a flow cytometer (BD Bioscience). Data were analysed using FlowJo software (TreeStar).

## Statistical analysis

Statistical analyses were performed using GraphPad Prism 6. Statistical comparisons were analysed using two-tailed unpaired Student's $t$-test.

## Reporting summary

Further information on research design is available in the Nature Portfolio Reporting Summary linked to this article.

## Data availability

All data supporting the results of this study are available within the paper and its Supplementary Information. The FACS data have been deposited in Flow Repository under the https://flowrepository.org/id/RvFrHHdN6hMvrfQgxfqUyuP3Y5TClVZtgoOL7neanuBhBTHcU8XFy0juOzfftSr7. Source data are provided with this paper.

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

## Acknowledgements

This study was funded by the Ministry of Science and Technology of the People's Republic of China (2021YFA1601400), the National Nature Science Foundation of China (22225603 and 22441051 to Z.L. and 82250710684 to Y.F.), the New Cornerstone Science Foundation (The XPLORER PRIZE) and Changping Laboratory. We also thank the facility support from the Analytical Instrumentation Centre of Peking University.

## Author contributions

Z.L. and Y.F. conceived the study. Z.D., assisted by Y.Z., F.Y., and Z.H., performed the chemical synthesis, analysis, and characterization. X.Y., assisted by Y.Z., H.G. and J.F., performed the immune-related assays and data analysis. Z.D., assisted by Y.Z., H.G. and J.F., performed cell assays, animal experiments, and data analysis. Z.D. and Y.Z. analysed the NMR spectra. Y.L. and Z.W performed the docking analysis, molecular simulation, and theoretical calculation. Z.D., assisted by Y.Z. and H.G., performed all other experiments. Z.D., X.Y., Y.Z., S.Q. and Q.S. analysed the data. Z.D., Z.L. and Y.F. wrote the manuscript with inputs from all authors. All authors discussed the results and commented on the manuscript.

## Competing interests

Z.L., Y.F., and Z.D. are co-inventors on a relevant patent application (PCT/CN2024/132467) filed by Changping Laboratory. Z.L. is a co-founder of and scientific advisor for BoomRay Pharmaceuticals. The remaining authors declare no competing interests.
