## [Transparent Peer Review file · Nature Communications]

Single Atom Engineering for Radiotherapy-Activated Immune Agonist Prodrugs

Corresponding Author: Professor Zhibo Liu

Version 0:

Reviewer comments:

Reviewer #1

(Remarks to the Author)

The authors describe a biotechnologically constructed novel radiotherapy-activated prodrug with inducing immunological effects within the tumour as well as abscopal effects outside the irradiated region.

Generally, the data have novelty and the method bears promise for further development. As a radiation oncologist with radiobiological knowlesge, I can only judge this part of the manuscript, not the bioengineering part: The radiobiological experiments are stringent as a proof of concept for the efficacy of the drug. For the animal experiments, I cannot find information about the size of the groups in the methods section. This should be added. as a comment for later experiments: It is preferable to have a longer follow-up and wait until you have recurrences also in the "best" treatment group. Then calculate the time until tumours reache.g. 2folg or 5fold starting volume. This omits problems of heterogeneous growth when you artificially decide for one follow-up timepoint and calculate the difference between the volumes at this one timepoint. This cannot be changed any more for the present analysis but should be ket in mind when the treatment shall be further developed.

Reviewer #2

(Remarks to the Author)

This is an interesting and impactful paper that uses a multidisciplinary approach to a major challenge in cancer immunotherapy. Can immunotherapy offers a huge amount of promise, but its realisation has been challenging. Driving immune activation using systemic delivery opens up major clinically limiting applicability when weighed against patient toxicity. The use of prodrugs and activation at the tumour site has been reported, but this paper offers a unique insight into applying modifications to immunotherapy drugs that allow them to be benign in the general circulation but be activated at the tumour site by the oxygen stripping ability of radiation. Some points:

- 1) I cannot comment on the chemistry but the approach and results would suggest that this was an excellent approach. Significant changes to the EC50 were profound and in vivo related toxicity much reduced.
- 2) Importantly, radiation caused activation, tumour killing and a bystander effect as outlined in the experiments. If this could be replicated at scale in humans then this could have a major impact on treatments using immunotherapy.

Some questions for clarification:

- 1) When the modification is made, are the resultant ,molecules stable? and for how long? as this could limit clinical applicability
- 2) Some comment on dosing and radiation doing at scale up could be mentioned, and what the limitations may be.

The paper is well written and the figures show significant promise.

Reviewer #3

(Remarks to the Author)

The authors report a new radiation-activated prodrug design of the TLR7/8 agonist R848. The concept of using local radiation to activate TLR signaling in tumor tissue is attractive, since systemically-administered TLR agonists can show toxicity, and immunosuppression in the irradiated tumor microenvironment may promote disease progression. The report

builds on prior development of N-oxide prodrugs for radiation-mediated drug activation — including of a TLR agonist that the group reported in ref. 40. In this present report, not shown in their prior study, the authors show convincing TLR agonist efficacy data in multiple mouse models, show evidence of cellular mechanisms in cell culture, and show the generalizability of the approach with other drugs (although, again, N-oxide approach was already reported by this group).

Addressing the following comments may improve the report:

I am not sure what “computer-aided single atom engineering” adds to the report. The authors have made an N-oxide prodrug of resiquimod that is very similar to the N-oxide prodrug of the very closely related molecule imiquimod that they reported in ref. 40. “SAE-RAP” seems to be a re-branding of what the group already did? It is not clear how computational modeling helped them arrive at something that was not already apparent since their results are so close to the prior reported N-oxide imiquimod. Why wasn’t that prior report ref. 40 not cited with proper context? It is currently the last cited manuscript in the paper, but should be one of the first.

The authors failed to cite Sun et al., “Radiation-Activated Resiquimod Prodrug Nanomaterials for Enhancing Immune Checkpoint Inhibitor Therapy”, which presents a radiation-activated R848 for cancer therapy. This report impacts the novelty of the present manuscript, since the concept, experimental demonstration in mice, and chemical strategy have significant overlap.

The authors in Sun et al. use a nanoparticle for their radiation-activated R848 perhaps in part because unencapsulated R848 has non-ideal pharmacokinetic properties, including relatively fast washout from tumor tissue. Can the authors shed light on this?

There is little data on the stability of the compounds, especially in mice. Weight loss is observed suggesting some activity of unirradiated prodrug. Where is that toxicity coming from? What is the long-term stability? The authors show blood counts but that does not seem to correlate with the weight loss toxicity observed from free R848 — in other words, what is the relevant tox marker to be looking at?

The report would be stronger if the authors provided more drug biodistribution, especially following radiation treatment. This is especially the case since prior reports have shown R848 prodrug can be reduced in the absence of radiation, albeit with a different chemical strategy (Sun et al *Adv. Mater.* 2023, 35, 2207733), and liver accumulation is potentially an issue.

It would be interesting to know how the N-oxide approach impacts cell permeability and brain accumulation. Any ability to accumulate in brain mets would be a huge advance and a major benefit over other reports in the literature.

The authors should expand discussion and analysis of human versus mouse TLR7 and TLR8. How is stability and activity in human cells?

The figures and statistical analyses are well presented. The text is clearly written. The manuscript is overall good, and the data are useful and informative for a diverse readership. However, this reviewer is left a bit confused over what the main message is and what the novelty is over prior reports.

Version 1:

Reviewer comments:

Reviewer #1

(Remarks to the Author)

My comments have been considered adequately

Reviewer #2

(Remarks to the Author)

The authors have answered all of my concerns in their rebuttal and I am happy that it be considered for publication by the editor.

Reviewer #3

(Remarks to the Author)

Thank you for addressing reviewer comments.

Responses to reviewers' comments:

Dear reviewers,

On behalf of the authorship group, I would like to thank you for taking the time to evaluate our manuscript. We are deeply grateful for your appreciation of our work. Please kindly find our point-by-point response to the reviewers' comments below. All the panels in Figures for Response have been edited into the appropriate section of the article or SI.

Reviewer 1

The authors describe a biotechnologically constructed novel radiotherapy-activated prodrug with inducing immunological effects within the tumour as well as abscopal effects outside the irradiated region. Generally, the data have novelty and the method bears promise for further development. As a radiation oncologist with radiobiological knowledge, I can only judge this part of the manuscript, not the bioengineering part: The radiobiological experiments are stringent as a proof of concept for the efficacy of the drug.

- We thank the reviewer for his/her enthusiastic comments on the novelty and promise for further development of our work.

1) For the animal experiments, I cannot find information about the size of the groups in the methods section. This should be added.

- As suggested, the size of the groups ($n = 6$ for treatment, and $n = 5$ for additional DMPK assay) of animal experiments has been added in the methods section in the revised version.

2) as a comment for later experiments: It is preferable to have a longer follow-up and wait until you have recurrences also in the "best" treatment group. Then calculate the time until tumours reach e.g. 2fold or 5fold starting volume. This omits problems of heterogeneous growth when you artificially decide for one follow-up timepoint and calculate the difference between the volumes at this one timepoint. This cannot be changed any more for the present analysis but should be kept in mind when the treatment shall be further developed.

- We thank the reviewer for the constructive suggestion. We agree with the point of reviewer that extending the follow-up period and waiting for recurrences in the "best" treatment group would provide a more accurate assessment of tumor growth. Also, calculating the time until tumors reach a certain fold increase in volume is indeed a better strategy to avoid the issues associated with heterogeneous. We will certainly keep this in mind for future experiments.

Reviewer 2

This is an interesting and impactful paper that uses a multidisciplinary approach to a

major challenge in cancer immunotherapy. Can immunotherapy offers a huge amount of promise, but its realisation has been challenging. Driving immune activation using systemic delivery opens up major clinically limiting applicability when weighed against patient toxicity. The use of prodrugs and activation at the tumour site has been reported, but this paper offers a unique insight into applying modifications to immunotherapy drugs that allow them to be benign in the general circulation but be activated at the tumour site by the oxygen stripping ability of radiation.

- We thank the reviewer's appreciation that this work is interesting and impactful.

Some points:

1) I cannot comment on the chemistry but the approach and results would suggest that this was an excellent approach. Significant changes to the EC50 were profound and *in vivo* related toxicity much reduced.

2) Importantly, radiation caused activation, tumour killing and a bystander effect as outlined in the experiments. If this could be replicated at scale in humans then this could have a major impact on treatments using immunotherapy.

The paper is well written and the figures show significant promise.

Some questions for clarification:

1) When the modification is made, are the resultant molecules stable? and for how long? as this could limit clinical applicability.

- We thank the reviewer for raising this question. Firstly, we test the storage stability of the *O*-R848 (solid powder). After being sealed and stored at -80 °C for about one year, the stability remained above 99.5%. In addition, we also tested the stability of *O*-R848 in human plasma over 2 hours at 37 °C and the results are shown in the **Fig. R1** (this data has been added in the revised version of supplementary information).

Fig. R1. Stability of *O*-R848 in human plasma in 2 hours at 37 °C.

Thus, we believe that *O*-R848 has sufficient stability for the experiments involved in this study. For future clinical applications, we will continue to test the storage stability and *in vivo* stability of such molecules.

2) Some comment on dosing and radiation doing at scale up could be mentioned, and what the limitations may be.

- We thank the reviewer for raising this question. In our view, a potential limitation is the rapid clearance of the small-molecule prodrug, which may make it difficult to align

with flexible clinical radiation protocols. We intend to address this issue with a potential covalent targeted prodrug strategy, which have been used in radionuclide therapy¹. If this strategy can enhance drug retention, it is expected to further reduce the dosing of drug administration and the dose required for radiotherapy.

Reviewer 3

The authors report a new radiation-activated prodrug design of the TLR7/8 agonist R848. The concept of using local radiation to activate TLR signaling in tumor tissue is attractive, since systemically-administered TLR agonists can show toxicity, and immunosuppression in the irradiated tumor microenvironment may promote disease progression. The report builds on prior development of N-oxide prodrugs for radiation-mediated drug activation — including of a TLR agonist that the group reported in ref. 40. In this present report, not shown in their prior study, the authors show convincing TLR agonist efficacy data in multiple mouse models, show evidence of cellular mechanisms in cell culture, and show the generalizability of the approach with other drugs.

The figures and statistical analyses are well presented. The text is clearly written. The manuscript is overall good, and the data are useful and informative for a diverse readership. However, this reviewer is left a bit confused over what the main message is and what the novelty is over prior reports.

Addressing the following comments may improve the report:

1) I am not sure what “computer-aided single atom engineering” adds to the report. The authors have made an N-oxide prodrug of resiquimod that is very similar to the N-oxide prodrug of the very closely related molecule imiquimod that they reported in ref. 40. “SAE-RAP” seems to be a re-branding of what the group already did? It is not clear how computational modeling helped them arrive at something that was not already apparent since their results are so close to the prior reported N-oxide imiquimod. Why wasn't that prior report ref. 40 not cited with proper context? It is currently the last cited manuscript in the paper, but should be one of the first.

- We thank the reviewer to point this out. Unlike our previous chemical discovery that radiotherapy can reduce *N*-oxides *in vivo*, (*J. Am. Chem. Soc.* 2022², 144, 9458–9464, which is cited in the introduction part of the revised version now), in this work we focused on developing a molecular engineering technique for prodrugs. We found a single oxygen atom can block the activity of drug molecules such as TLR7/8 agonists, leading to the development of the “SAE-RAP” molecular engineering technique, which has not been reported previously. We agree with the reviewer about the phrase “computer-aided” and it has been removed from the title in the revised version, as computer-aided drug design is not the central focus of this work and docking analysis was used to suggest that the “SAE-RAP” technique may have broad applicability for RAPs.

2): The authors failed to cite Sun et al., “Radiation-Activated Resiquimod Prodrug Nanomaterials for Enhancing Immune Checkpoint Inhibitor Therapy”, which presents

a radiation-activated R848 for cancer therapy. This report impacts the novelty of the present manuscript, since the concept, experimental demonstration in mice, and chemical strategy have significant overlap.

- We thank the reviewer for raising this question. Sun et al. reported R848-N₃ based nanomedicines have certainly contributed to the development of radiotherapy-activated prodrugs. We have cited their relevant publications in the revised version. In this work, we aim to investigate a molecular engineering approach specifically for small molecule prodrugs. In addition, the radiation-responsive chemical processes and mechanisms of the two prodrug systems are distinctly different. Therefore, we believe our work and that of Sun et al. represent different directions in the development of the RAPs.

3): The authors in Sun et al. use a nanoparticle for their radiation-activated R848 perhaps in part because unencapsulated R848 has non-ideal pharmacokinetic properties, including relatively fast washout from tumor tissue. Can the authors shed light on this?

- We thank the reviewer for the constructive suggestion. Indeed, it has been reported that small molecule TLR7/8 agonists such as R848 face issues like rapid clearance³. However, we believe that the exceptionally fast rate of radiation-induced deoxygenation (**Fig. R2** and **R3**) is well aligned with the short metabolic half-life of these small molecule prodrugs, enabling them to be effectively activated during radiotherapy when the drug concentration in the tumor tissue reaches its peak (**Fig. R4**), thus avoiding ineffective activation before renal clearance. Importantly, we consider the integration of nanotechnology to improve the pharmacokinetic properties of prodrugs as a highly promising strategy, which could potentially be one of the key directions for future development for RAPs.

Fig. R2. Time-dependent release of R848 from *O*-R848 (10 μM in PBS solution) irradiated by 60 Gy X-ray detected by UPLC-MS.

Fig. R3. Time-dependent fluorescence intensity change (a) and photographs (b) of *O*-coumarin after 60 Gy radiation.

Fig. R4. **a**, Time-dependent accumulation of *O*-R848 in blood and tumour detected by UPLC-MS ($n = 3$ mice). **b**, Concentration of R848 released in tumours treated with or without 6 Gy X-ray irradiation at 1 h after intravenously administered of *O*-R848 (30 $\mu\text{mol/kg}$), detected by UPLC-MS ($n = 5$ mice).

4): There is little data on the stability of the compounds, especially in mice. Weight loss is observed suggesting some activity of unirradiated prodrug. Where is that toxicity coming from? What is the long-term stability? The authors show blood counts but that does not seem to correlate with the weight loss toxicity observed from free R848 — in other words, what is the relevant tox marker to be looking at?

- We thank the reviewer for raising this question. Firstly, we tested the storage stability of the *O*-R848 (solid powder). After being sealed and stored at $-80\text{ }^{\circ}\text{C}$ for about one year, the stability remained above 99.5%. In addition, we also tested the time-dependent human plasma stability of *O*-R848, and release of R848 in blood, tumour, liver, kidney, and brain tissues without radiotherapy treated. These results are shown in the **Fig. R5**, which have been added in the revised version of supplementary information.

Fig. R5. **a**, Stability of *O*-R848 in human plasma in 2 hours at $37\text{ }^{\circ}\text{C}$. R848 release in blood (**b**), tumour (**c**), liver (**d**), kidney(**e**), and brain (**f**) without radiotherapy treated, detected by UPLC-MS, $n = 5$.

As shown in **Fig. R6a**, compared to R848, a 3-fold dose of *O*-R848 displayed smaller changes in body weight and no mortality in mice after 3 days of continuous iv. injection.

Additionally, the maximum weight loss remained below the termination threshold (over 20% weight loss). Moreover, in our subsequent work, we chose the safer dose of 30 $\mu\text{mol/kg}$ (with a maximum weight loss of less than 5%, which can also recover). As shown in **Fig. R6b**, *O*-R848 prevented the excessive cytokine production induced by R848. On the other hand, the weight changes may also be influenced by the administration process and the exogenous prodrug molecules, which could affect factors such as energy metabolism, appetite, or water-salt balance, rather than indicating direct on-target toxicity. These reactions may result in several changes in peripheral blood cells, including alterations in white blood cells and lymphocytes.

Fig. R6. a. Body weight change measurements of C57BL/6J mice ($n = 6$ for each group) following 3 times intravenous administration of R848 and *O*-R848, respectively. **b.** Serum IFN- γ and TNF- α measurements taken from C57BL/6J mice at 4 h post-intravenous injection of R848 and *O*-R848 at a dose of 30 $\mu\text{mol/kg}$, respectively. $n = 3$ for each group tested.

Fig. R7. Complete blood count analysis. Blood samples were collected at a time point of 7 days post-treatments.

As shown in **Fig. R7**, R848 activates leukocytes and promotes the differentiation of monocytes into dendritic cells, which secrete cytokines such as IL-1, IL-6, and TNF- α . These cytokines further drive the immune response, the number of leukocytes may sharply increase, resulting in leukocytosis in peripheral blood. Overactivated leukocytes exacerbate tissue damage and can even lead to organ failure. The immune-stimulating effects of R848 not only activate the innate immune system but also enhance adaptive immune responses, particularly through the activation and proliferation of T cells, leading to a marked increase in lymphocyte count and further amplifying the excessive systemic immune response.

5): The report would be stronger if the authors provided more drug biodistribution, especially following radiation treatment. This is especially the case since prior reports have shown R848 prodrug can be reduced in the absence of radiation, albeit with a different chemical strategy (Sun et al Adv. Mater. 2023, 35, 2207733), and liver accumulation is potentially an issue.

- We thank the reviewer for the constructive suggestion. In addition to the existing time-dependent distribution in blood and tumor, we have also included the time-dependent distribution of the drug in the liver and kidney. As shown in **Fig. R8**, the small molecule prodrug is primarily metabolized by the kidneys, with rapid clearance in both the liver and kidneys (cleared within 3 hours).

Fig. R8. Time-dependent accumulation of *O*-R848 in liver (a) and kidney (b), detected by UPLC-MS ($n = 5$ mice).

Furthermore, we also assessed the time-dependent distribution of R848 produced after irradiation (**Fig. R9**). We found after the generation of R848 in tumor, R848 exhibits a time-dependent distribution mainly in tumours. However, generated R848 also exhibits blood and kidney distribution for less than 2 hours.

Fig. R9. Time-dependent distribution of released R848 from tumours after radiotherapy treated, detected by UPLC-MS, $n = 5$.

These results have been added in the revised version of supplementary information.

6): It would be interesting to know how the N-oxide approach impacts cell permeability and brain accumulation. Any ability to accumulate in brain mets would be a huge advance and a major benefit over other reports in the literature.

- We thank the reviewer for raising this question. Through parallel artificial membrane permeability assay (PAMPA), the effective permeability coefficients (P_e) of R848 and *O*-R848 are 14.62 ± 1.31 and 0.43 ± 0.09 . This result means cell permeability of R848 was ~ 34 -fold reduced via single oxygen atom engineering. We also tested time-dependent accumulation of *O*-R848 in brain tissues (**Fig. R10**). We appreciate the reviewer's constructive suggestion, which provides valuable insights for the future development of RAP technique for treating brain-relevant tumours. However, we think the BBB permeability of such prodrugs needs further development to achieve greater activation efficacy. These results have been added in the revised version of supplementary information.

Fig. R10. Time-dependent accumulation of *O*-R848 in brain, detected by UPLC-MS ($n = 5$ mice).

7): The authors should expand discussion and analysis of human versus mouse TLR7 and TLR8. How is stability and activity in human cells?

- We thank the reviewer for raising this question. As shown in **Fig. R11**, we tested the TLR8 activation of R848, *O*-R848 and *O*-R848 treated with X-ray in human TLR8 reporter HEK293 cells. Compared to R848, *O*-R848 can effectively reduce the immunostimulatory activity even in human-derived reporter cells. *O*-R848's immunostimulatory activity can be effectively restored after X-ray irradiation. These results have been added in the revised version of supplementary information.

Fig. R11. Human TLR8 activation of R848 (10 µM), *O*-R848 (10 µM), and X-ray treated *O*-R848 (10 µM) in hTLR8 report cells after 24 h incubation ($n = 3$ for each group). Data are presented as mean values \pm s.d.

Reference

1. Cui, X.-Y. et al. Covalent targeted radioligands potentiate radionuclide therapy. *Nature* **630**, 206-213 (2024).
2. Ding, Z. et al. Radiotherapy Reduces N-Oxides for Prodrug Activation in Tumors. *Journal of the American Chemical Society* **144**, 9458-9464 (2022).
3. Bhagchandani, S., Johnson, J.A. & Irvine, D.J. Evolution of Toll-like receptor 7/8 agonist therapeutics and their delivery approaches: From antiviral formulations to vaccine adjuvants. *Advanced Drug Delivery Reviews* **175**, 113803 (2021).

Responses to reviewers' comments:

Reviewer 1

My comments have been considered adequately.

- We thank the reviewer again for taking the time to evaluate our manuscript.

Reviewer 2

The authors have answered all of my concerns in their rebuttal and I am happy that it be considered for publication by the editor.

- We thank the reviewer again for taking the time to evaluate our manuscript.

Reviewer 3

Thank you for addressing reviewer comments.

- We thank the reviewer again for taking the time to evaluate our manuscript.